# Efficient Over-parameterized Matrix Sensing via Alternating Preconditioned Gradient Descent

## Abstract

We consider solving the low-rank matrix sensing problem in the over-parameterized setting, where the specified rank is larger than the true rank. Precisely, our main objective is to recover a matrix $X^* \in \mathbb{R}^{n_1 \times n_2}$ with rank $r_\star$ using an over-parameterized form $LR^\top$, where $L \in \mathbb{R}^{n_1 \times r}$, $R \in \mathbb{R}^{n_2 \times r}$ and $\min\{n_1, n_2\} \geq r > r_\star$ with the true rank $r_\star$ being unknown. The commonly used methods tackling such a problem such as Factorized Gradient Descent (FGD) can only demonstrate sub-linear convergence behavior, and their performance could significantly deteriorate when the matrix condition number is relatively large. To address this issue, we propose the alternating preconditioned gradient descent (APGD) method that an inexpensive right preconditioner with a constant damping parameter is applied to the original gradient. We prove that even starting from a random initialization, APGD can recover the target matrix at a linear convergence rate in the over-parameterized situation, independent of the condition number. Notably, unlike previous FGD-based methods, APGD alternates between updating the two factor matrices, which eliminates the reliance on a small step size, thereby enabling faster convergence. Through a series of experiments, we demonstrate that APGD achieves the fastest convergence speed compared to other methods, and further possesses strong robustness with respect to step size, condition number and other parameters.

## 1 Introduction

Low-rank matrix sensing is a fundamental problem encountered in various fields, including image processing (Candès et al., 2011; Li et al., 2019; Arora et al., 2019), phase retrieval (Vaswani et al., 2017; Nayer & Vaswani, 2021), quantum tomography (Rambach et al., 2021), etc. The primary objective is to recover a rank $r_\star$ matrix $X_\star \in \mathbb{R}^{n_1 \times n_2} (r_\star \ll \min\{n_1, n_2\})$ from linear measurements $\{(y_i, A_i)\}_{i=1}^m$ of the form

$$y_i = \langle A_i, X_\star \rangle, i = 1, ..., m. \tag{1}$$

This model can be concisely expressed as $y = \mathcal{A}(X_\star)$, where $\mathcal{A}(X_\star) = [\langle A_1, X_\star \rangle, \langle A_2, X_\star \rangle, \cdots, \langle A_m, X_\star \rangle]$ is the so-called measurement operator. A prevalent method for recovering a low-rank matrix $X \in \mathbb{R}^{n_1 \times n_2}$ involves solving the following problem:

$$\arg\min \text{rank}(X) \text{ s.t. } y = \mathcal{A}(X). \tag{2}$$

However, such an optimization problem is NP-hard due to the nonconvex rank constraint. To address this challenge, researchers have proposed relaxing the rank constraint to a convex nuclear norm constraint (Recht et al., 2010; Candes & Plan, 2011; Candes & Recht, 2012; Candès & Tao, 2010). Although this kind of convex relaxation approach provides a tractable solution, it fails to fully exploit the low-rankness of $X_\star$, and as a result, the computational cost significantly increases as the matrix size grows. To mitigate the computational overhead, a common approach is to decompose the matrix $X$ into a factorized form $LR^\top$, where $L \in \mathbb{R}^{n_1 \times r}, R \in \mathbb{R}^{n_2 \times r}$ and $r$ is the estimated rank, also known as the Burer-Monteiro method (Burer & Monteiro, 2003; 2005), and then solve

$$\arg\min_{L, R} f(L, R) = \frac{1}{2}\|\mathcal{A}(LR^\top) - y\|_2^2. \tag{3}$$

This problem can be efficiently solved by the widely used factorized gradient descent method (Tu et al., 2016; Zhuo et al., 2021; Jin et al., 2023; Xiong et al., 2024).

$$L_{t+1} = L_t - \eta \nabla_L f(L_t, R_t), \quad R_{t+1} = R_t - \eta \nabla_R f(L_t, R_t), \eta \text{ denotes the step-size}.$$

However, the following challenges remain in practice:

- **ill-conditioning** It is well known that gradient methods are susceptible to the condition number $\kappa$ of a matrix, defined as the ratio of the largest to the smallest singular value. The number of gradient descent iterations increases at least linearly with the condition number (Zheng & Lafferty, 2015). Unfortunately, most practical datasets exhibit very large condition numbers, e.g., (Cloninger et al., 2014) notes that certain applications of matrix sensing have condition numbers as high as $\kappa = 10^{15}$.

- **Over-parameterization** A major challenge is the lack of prior information about the true rank $r_\star$. Therefore, in the Burer-Monteiro method, we typically set $r_\star \leq r \leq \min\{n_1, n_2\}$, a scenario referred to as over-parameterization. However, recent studies (Zhuo et al., 2021; Zhang et al., 2021; Xiong et al., 2024) have shown that in the case of over-parameterization, the convergence behavior of factorized gradient methods can be significantly affected.

- **Initialization** For factorized gradient descent method, obtaining an initial point is the first step. Many previous studies (Zhuo et al., 2021; Tong et al., 2021; Zhang et al., 2021) rely on spectral initialization to obtain a good initial point that is close to the true solution $X_\star$. However, recent researches (Stöger & Soltanolkotabi, 2021; Lee & Stöger, 2023; Xiong et al., 2024) have shown that small random initialization can achieve similar results to spectral initialization. Nevertheless, small initialization typically requires a larger number of iterations and is less practical in some big data applications.

Given these challenges, the main goal of this work is to address the following question: **Can one develop an efficient and robust method for solving ill-conditioned matrix sensing in the over-parameterized setting at a linear convergence rate with a proper random initialization ?**

## 1.1 OUR CONTRIBUTION

To answer the aforementioned question, we consider using an alternating preconditioned gradient descent (APGD) method to solve the over-parameterized matrix sensing problem. Preconditioning is a commonly used method to address issues related to uneven distributions of matrix singular values (Nocedal & Wright, 1999; Carr et al., 2021; Zheng et al., 2021). Recently, this approach has also been applied to matrix sensing problems (Tong et al., 2021; Zhang et al., 2021; Xu et al., 2023). However, existing preconditioned gradient descent methods are mostly tailored for symmetric positive definite matrices, which is not a very practical assumption.

To this end, we consider in this work recovering arbitrary matrices. After decomposing the non-symmetric matrix into two factor matrices, a natural idea is to update these two matrices alternately (Tanner & Wei, 2016; Ward & Kolda, 2023; Jia et al., 2024). Compared to the vanilla gradient descent, the advantage of alternating gradient descent is that it allows for larger step sizes, thus speeding up convergence. As a result, we developed the alternating preconditioned gradient descent

---

**Algorithm 1** Solving (3) by alternating preconditioned gradient descent (APGD)

---

**Input:** Observation $\{y_i, \mathcal{A}_i\}_{i=1}^m$, step size $\eta$, estimated rank $r$, initialization scale $c_1$, damping parameter $\alpha$

**Initialization**: Let $L_0 = \frac{c_1}{3\sqrt{n_1+r}} \widetilde{L}_0 \in \mathbb{R}^{n_1 \times r}$, $R_0 = \frac{c_1}{3\sqrt{n_2+r}} \widetilde{R}_0 \in \mathbb{R}^{n_2 \times r}$, where the entries of $\widetilde{L}_0, \widetilde{R}_0$ are i.i.d. Gaussian entries with distribution $\mathcal{N}(0,1)$

1: **for** $t = 0$ to $T - 1$ **do**
2:     $L_{t+1} = L_t - \eta \nabla_L f(L_t, R_t) \cdot (R_t^\top R_t + \alpha I)^{-1}$
3:     $R_{t+1} = R_t - \eta \nabla_R f(L_{t+1}, R_t) \cdot (L_{t+1}^\top L_{t+1} + \alpha I)^{-1}$
4: **end for**
5: **return:** $X_T = L_T R_T^\top$

---

algorithm for solving (3), as shown in Algorithm 1. APGD starts from a randomly initialized point that is not too small and alternately updates the two factor matrices $L_t$ and $R_t$. In each iteration, APGD applies a right preconditioner to the original gradient, thereby accelerating the convergence. Moreover, initialization is a critical aspect of optimization algorithms. Previous methods largely relied on spectral initialization or extremely small random initialization. We are the first to prove that with an easy-to-use random initialization, APGD can solve the asymmetric over-parameterization matrix sensing problem at a linear convergence rate.

**Theorem 1.** *(Informal) In the over-parameterization $(r \geq r_\star)$ situation, under some mild assumptions, starting from a random initialization which is not too small, APGD achieves a $\epsilon$ accuracy minima, i.e. $\|L_t R_t - X_\star\|_F \leq \epsilon$, with $\Omega(\log \frac{r}{\epsilon})$ iterations.*

We summarize our contributions as follows:

- Firstly, we propose a preconditioned alternating gradient descent algorithm, which can converge from a random initial point to the true solution at a linear rate. Even in cases with a large condition number and severe rank overestimation, APGD achieves the fastest convergence. Compared to previous methods, APGD maintains convergence even with large step sizes, offering a faster convergence rate. Moreover, unlike other globally convergent algorithms, APGD does not rely on extremely small initialization, thus reducing the number of iterations.

- Secondly, we develop a two-phase analytical framework that divides the convergence of APGD into an initial phase and a local convergence phase. Using this framework, we analyze the convergence rate of APGD under over-parameterization and random initialization, demonstrating that APGD exhibits linear convergence. We believe this framework can also be extended to other tasks such as matrix completion, one-bit matrix sensing, and phase retrieval.

- Thirdly, we conduct a series of experiments, showing that APGD achieves the fastest convergence speed compared to other methods. Additionally, we perform sensitivity tests on APGD's parameters, such as step size, and damping parameter, initialization scale, demonstrating that APGD exhibits strong robustness to parameter variations.

Table 1: Comparison of related works. The second column indicates whether the over-rank case is considered. In the third column, 'local' refers to initial points very close to the ground truth, 'small random' refers to initial values with a very small scale, and 'random' refers to initial values with a scale comparable to the ground truth. The fourth column shows the number of iterations required for the algorithm to converge to an $\epsilon$-global minima, where $\kappa$ represents the condition number. The fifth column indicates whether the asymmetric factorization is considered.

| methods | over rank | init. | iteration complexity | asymmetry |
|---|---|---|---|---|
| (Tong et al., 2021) | ✗ | local | $\log \frac{1}{\epsilon}$ | ✓ |
| (Zhuo et al., 2021) | ✓ | local | $\frac{1}{\epsilon}$ | ✗ |
| (Zhang et al., 2021) | ✓ | local | $\log \frac{1}{\epsilon}$ | ✗ |
| (Stöger & Soltanolkotabi, 2021) | ✓ | small random | $\kappa^8 + \kappa^6 \log(\frac{\kappa n}{\epsilon})$ | ✗ |
| (Xu et al., 2023) | ✓ | small random | $\log \kappa \cdot \log(\kappa n) + \log \frac{1}{\epsilon}$ | ✗ |
| (Xiong et al., 2024) | ✓ | small random | $\log \kappa + \log \frac{\kappa n}{\epsilon}$ | ✓ |
| (Lee & Stöger, 2023) | ✗ | random | $\log n + \log \frac{1}{\epsilon}$ | ✓ |
| ours | ✓ | random | $\log \frac{r}{\epsilon}$ | ✓ |

## 2 RELATED WORK

In recent years, a major research direction in the field of matrix sensing has been the development of fast and efficient non-convex algorithms, with the factorized gradient descent algorithm, particularly the Burer-Monteiro (BM) factorization (Tu et al., 2016; Zhuo et al., 2021; Chen & Wainwright, 2015; Sun & Luo, 2016), being a representative example. Despite the significant progress made

in the study of the FGD algorithm, it still performs poorly in cases of ill-conditioning and over-parameterization, which has led to extensive research efforts addressing these issues. Additionally, the initialization of FGD has become another prominent research focus. We present a comparison of several works most relevant to our approach in Table 1.

**ill-conditioning**
Gradient-based methods are highly sensitive to the condition number of matrices, and the iteration complexity of the FGD algorithm grows linearly with the matrix condition number, i.e. $\mathcal{O}(\kappa \log 1/\epsilon)$, as the condition number increases, the convergence rate of FGD significantly slows downZheng & Lafferty (2015); Zhang et al. (2023). In recent years, there has been a series of studies focused on addressing this problem using preconditioning methods (Mishra et al., 2012; Wei et al., 2016; Mishra & Sepulchre, 2016; Tanner & Wei, 2016; Tong et al., 2021; Zhang et al., 2021; 2023; 2022; Bian et al., 2023). Most of these works rely on a good initial point and only conduct local convergence analysis.

**Over-parameterization** In earlier years, a series of works (Tu et al., 2016; Tong et al., 2021; Chen & Wainwright, 2015; Li et al., 2018) demonstrated that under the exact rank assumption, the factorized gradient descent method could converge to the ground truth at a linear rate. However, since it is difficult to obtain the rank of the matrix to be recovered in practice, recent research has focused on matrix recovery in the overestimated rank setting (Zhuo et al., 2021; Li et al., 2018; Stöger & Soltanolkotabi, 2021; Soltanolkotabi et al., 2023). However, over-parameterization exacerbates the ill-conditioning of the problem, leading to slower convergence rates. Studies by Zhang et al. (2021; 2023); Xu et al. (2023); Cheng & Zhao (2024) have investigated the issue of slow convergence in over-parameterized settings.

**Initialization**
Early methods demonstrated that, starting from an initial point obtained through spectral initialization (Chen & Wainwright, 2015; Sun & Luo, 2016), which is close to the ground truth, the factorization gradient descent algorithm can converge to the optimal solution. In the past two to three years, some studies (Bhojanapalli et al., 2016; Zhang et al., 2019; Ge et al., 2016; 2017; Zhu et al., 2021) have shown that, under certain conditions, all local minima of the low-rank matrix sensing problem are also global minima. Consequently, global convergence with random initialization has become a research focus Jin et al. (2023); Ding et al. (2022); Jiang et al. (2023); Soltanolkotabi et al. (2023); Chen et al.. (Stöger & Soltanolkotabi, 2021) revealed that in the noiseless case, gradient descent with small random initialization performs similarly to spectral initialization.

## 3 MAIN RESULTS

### 3.1 PRELIMINARIES

**Notations** Singular values of a rank-$r$ matrix $X$ are donated as $\|X\| = \sigma_1(X) \geq \sigma_2(X) \geq \cdots \geq \sigma_r(X) > 0$. We denote the condition number of $X$ as $\kappa(X) = \sigma_1(X)/\sigma_r(X)$.

**Definition 3.1.** *(Restricted Isometry Property) The linear map $\mathcal{A}$ is said to satisfies Restricted Isometry Property (RIP) with parameters $(r, \delta_r)$ if there exits constants $0 \leq \delta_r < 1$ and $m > 0$ such that for every rank-$r$ matrix $M$, it holds that*

$$(1 - \delta_r)\|M\|_F^2 \leq \|\mathcal{A}(M)\|_2^2 \leq (1 + \delta_r)\|M\|_F^2.$$

**Lemma 1.** *If all the entries of the measurement matrices $\{A_i\}_{i=1}^m$ are (sub-)gaussian random variables with zero mean and variance $1/m$ and $m \geq D(n_1 + n_2)r$, then the linear map $\mathcal{A}$ satisfies the restricted isometry property of order $r$ with constant $\delta_r > 0$ with probability exceeding $1 - Ce^{-dm}$ for fixed constants $D, d > 0$ (Candes & Plan, 2011).*

**Assumption 1.** *(Assumption of initialization) Suppose that we sample $\widetilde{L_0} \in \mathbb{R}^{n_1 \times k}$, $\widetilde{R_0} \in \mathbb{R}^{n_2 \times k}$ with i.i.d. $N(0, \sigma_1(X_\star))$ entries. Then we take $L_0 = \frac{c_1}{3\sqrt{n_1+k}}\widetilde{L_0}$ and $R_0 = \frac{c_1}{3\sqrt{n_2+k}}\widetilde{R_0}$ with $c_1 \geq c_{init}$.*

Under this initialization, we only need to know the dimensions $n_1$, $n_2$ of the target matrix $X_\star$ and an overestimated rank $r \geq r_\star$; the true rank $r_\star$ of the matrix is not required. Similar random initialization methods have also been adopted in other works (Jiang et al., 2022), and it holds with high probability as proved in Appendix C.

**Assumption 2.** *(Assumption of the linear map $\mathcal{A}$) The linear map $\mathcal{A}$ satisfies the rank-$2r$ RIP with parameter $\delta_{2r} \leq \sqrt{2} - 1$.*

RIP is a commonly used condition in the field of compressed sensing, which states that the operator $\mathcal{A}(\cdot)$ approximately preserves the distances between low-rank matrices. It serves as a bridge between fully observed and partially observed data. We can first analyze the population case and then extend the results to the sample case using the RIP. Assumption 2 ensures that $\sigma_{\min}(L_t)$ and $\sigma_{\min}(R_t)$ exhibit a linear convergence rate in the initial phase.

### 3.2 MAIN THEOREM

**Theorem 2.** *Assume that we have the assumption 1 and assumption 2 hold, and $0 < \eta < \frac{2}{1+\delta_{2r}}$, $\alpha = \mathcal{O}(\epsilon^2)$, then solving matrix sensing problem (3) with algorithm 1 leads to:*

$$f(L_{t+1}, R_{t+1}) \leq (1 - \eta_c)^2 f(L_t, R_t),$$

*and*

$$\|L_T R_T^\top - X_\star\|_F \leq (1 - \eta_c)^T \sqrt{\frac{1 + \delta_{2r}}{1 - \delta_{2r}}} \|L_0 R_0^\top - X_\star\|_F$$

*with $\eta_c = \eta(1 - \frac{1}{2}(1 + \delta_{2r})\eta\mu_P)$, $\mu_P = \frac{1-\delta_{2r}}{4}$.*

**Remark 3.1.** *Iteration complexity From Theorem 2, it is clear that the iteration complexity of APGD is independent of the condition number $\kappa$. Second, APGD is capable of larger step sizes, and the maximum step size is related to the RIP constant $\delta_{2r}$; the smaller the $\delta_{2r}$, the larger the range of step size.*

**Remark 3.2.** *Robustness to step size It is worth noting that the step size constraint here is $0 < \eta < \frac{2}{1+\delta_{2r}}$, which is quite a loose bound. This shows that our method is able to iterate with a larger step size and thus converge faster. Other gradient descent-based methods typically require much smaller step sizes, resulting in slower convergence. In Zhang et al. (2021), the step size is restricted as $\eta \leq \frac{1}{16+C_{lb}}$; in Tong et al. (2021), the step size is restricted as $0 < \eta < 2/3$; in Zhuo et al. (2021), the step size is set as $\eta = \frac{1}{100\sigma_1}$; in Xu et al. (2023), the step size is restricted as $0 < \eta \leq c_\eta$, where $c_\eta$ is some sufficiently small constant.*

**Remark 3.3.** *Robustness to damping parameter Note that we only require the damping parameter $\alpha = \mathcal{O}(\epsilon^2)$, which is a quite loose upper bound. This makes APGD highly robust to the damping parameter. In contrast, other methods impose stricter requirements on the range of values for the damping parameter. For example, the PrecGD algorithm (Zhang et al., 2021) requires $C_{lb}\|L_t R_t^\top - X_\star\|_F \leq \alpha_t \leq C_{ub}\|L_t R_t^\top - X_\star\|_F$ and necessitates dynamic updates of $\alpha_t$ during iterations. Similarly, the ScaledGD($\lambda$) method proposed by Xu et al. (2023) requires $0.01c_\alpha\sigma_{\min}^2(X_\star) \leq \alpha \leq c_\alpha\sigma_{\min}^2(X_\star)$ with some sufficiently small $c_\alpha$, which is also a relatively strict constraint.*

## 4 PROOF SKETCH

### 4.1 SUB-LINEAR CONVERGENCE OF (ALTERNATING) GRADIENT DESCENT

It's proved by Zhang et al. (2021; 2023) that vanilla gradient descent convergence at a sub-linear rate of

$$f(X_{t+1}) \leq (1 - \eta\sigma_{\min}(X_t))f(X_t), \ \sigma_{\min}(X_t) = \sigma_r(X_t), \ r = \text{rank}(X_t)$$

when solving the over-parameterized matrix sensing problem. As $X$ converges to $X_\star$, $\sigma_{\min}(X)$ approaches 0, leading to a severe decrease in the rate of convergence.

As for the alternating gradient descent, it would be similarly affected. By direct algebraic calculations coupled with the $(2r, \delta_{2r})$-RIP condition, we get

$$
\begin{aligned}
f(L_{t+1}, R_t) &= \frac{1}{2}\|\mathcal{A}(L_{t+1}R_t^\top - X_\star)\|_2^2 \\
&= \frac{1}{2}\|\mathcal{A}([L_t - \eta\nabla_L f(L_t, R_t)]R_t^\top - X_\star)\|_2^2 \\
&\leq f(L_t, R_t) + 0.5\eta^2\|\mathcal{A}(\nabla_L f(L_t, R_t)R_t^\top)\|_2^2 - \eta\|\nabla_L f(L_t, R_t)\|_F^2 \\
&\leq f(L_t, R_t) - \eta\left[2 - \eta(1 + \delta_{2r})\sigma_1^2(R_t)\right](1 - \delta_{2r})\sigma_{\min}^2(R_t)f(L_t, R_t).
\end{aligned}
$$

(4)

Similarly, we have

$$f(L_{t+1}, R_{t+1}) \leq f(L_{t+1}, R_t) - \eta \left[ 2 - \eta(1 + \delta_{2r})\sigma_1^2(L_{t+1}) \right] (1 - \delta_{2r})\sigma_{\min}^2(L_{t+1}) f(L_{t+1}, R_t). \tag{5}$$

It can be observed that, similar to the vanilla gradient descent, as the $X$ converges to $X_\star$, $\sigma_{\min}(R)$ and $\sigma_{\min}(L)$ approach 0, leading to the sub-linear convergence.

To solve this problem, our approach is to add the right preconditioners to the original gradient, which acts as a sort of Newton's method. However, unlike Newton's method, instead of computing the inverse of the huge-sized Hessian matrix (which is of the size $(n_1 + n_2)r \times (n_1 + n_2)r$), we only need to compute the inverse of two $r \times r$ matrices, which greatly reduces the computational overhead. In the following, we will present how our method works, mainly by making the lower bound of the gradient's Frobenius norm independent of $\sigma_{\min}(L)$ and $\sigma_{\min}(R)$ via the preconditioner.

### 4.2 CONVERGENCE ANALYSIS

To begin with our analysis, we first rewrite equation (4) updated by APGD. Then we get the following lemma.

**Lemma 2.** *Suppose that the linear map $\mathcal{A}(\cdot)$ satisfy the RIP with parameters $(2r, \delta_{2r})$, considering the APGD in Algorithm 1, then we have*

$$f(L_{t+1}, R_t) \leq f(L_t, R_t) - \eta(1 - \frac{\eta}{2}(1 + \delta_{2r}))\|\nabla_L f(L_t, R_t)(R_t^\top R_t + \alpha I)^{-1/2}\|_F^2$$

$$f(L_{t+1}, R_{t+1}) \leq f(L_{t+1}, R_t) - \eta(1 - \frac{\eta}{2}(1 + \delta_{2r}))\|\nabla_R f(L_{t+1}, R_t)(L_{t+1}^\top L_{t+1} + \alpha I)^{-1/2}\|_F^2$$

Next we need to obtain an upper bound on the preconditioned gradient as

$$\|\nabla_L f(L_t, R_t)(R_t^\top R_t + \alpha I)^{-1/2}\|_F^2 \geq \mu_P f(L_t, R_t)$$
$$\|\nabla_R f(L_{t+1}, R_t)(L_{t+1}^\top L_{t+1} + \alpha I)^{-1/2}\|_F^2 \geq \mu_P f(L_{t+1}, R_t), \tag{6}$$

where $\mu_P$ is constant independent of $\sigma_{\min}(L)$ and $\sigma_{\min}(R)$, then we can obtain linear convergence

$$f(L_{t+1}, R_{t+1}) \leq (1 - \eta_c)^2 f(L_t, R_t), \ \eta_c = \eta(1 - \frac{\eta}{2}(1 + \delta_{2r}))\mu_P. \tag{7}$$

Based on the recovery error, we divide the convergence analysis of APGD into two stages, : **initial stage**, where $\|L_t R_t^\top - X_\star\|_F \geq \rho\sigma_{r_\star}(X_\star)$ and **local convergence stage** where $\|L_t R_t^\top - X_\star\|_F < \rho\sigma_{r_\star}(X_\star)$.

#### 4.2.1 STAGE 1

In the initial stage, the recovery error is relatively large and $L_t^\top L_t$, $R_t^\top R_t$ are non singular, which indicate the damping parameter $\alpha$ can be infinitesimal. Specifically, we can bound the preconditioned gradient as

$$\|\nabla_L f(L_t, R_t)(R_t^\top R_t + \alpha I)^{-1/2}\|_F^2 = \|\mathcal{A}^*\mathcal{A}(L_t R_t^\top - X_\star)R_t(R_t^\top R_t + \alpha I)^{-1/2}\|_F^2$$

$$\geq (1 - \delta_{2r})\sigma_{\min}^2 \left( R_t(R_t^\top R_t + \alpha I)^{-1/2} \right) \|L_t R_t^\top - X_\star\|_F^2$$

$$= \frac{2(1 - \delta_{2r})}{1 + \alpha/\sigma_{\min}^2(R_t)} f(L_t, R_t), \tag{8}$$

where $\mathcal{A}^*$ denotes the adjoint operator of $\mathcal{A}$. And $\|\nabla_R f(L_{t+1}, R_t)(L_{t+1}^\top L_{t+1} + \alpha I)^{-1/2}\|_F^2$ can be bounded similarly.

If we take a infinitesimal $\alpha$, then we have $\frac{2(1-\delta_{2r})}{1+\alpha/\sigma_{\min}^2(R_t)} \geq (1 - \delta_{2r})$. Based on this, we give the following lemma bounding the preconditoned gradient.

**Lemma 3.** *Assume that we have assumption 1 holds with $c_1 \geq c_{init} = \sqrt{\frac{2\alpha}{\sigma_1(X_\star)\sigma_{r_\star}^2(X_\star)}}$ and the linear map $\mathcal{A}$ satisfies the rank-2r RIP condition with $\delta_{2r} \leq \sqrt{2} - 1$, and then take $\alpha = \mathcal{O}(\epsilon^2)$,*

*where $\epsilon$ denotes the final recovery error, then we have*

$$\|\nabla_L f(L_t, R_t)(R_t^\top R_t + \alpha I)^{-1/2}\|_F^2 \geq (1 - \delta_{2r})f(L_t, R_t)$$
$$\|\nabla_R f(L_{t+1}, R_t)(L_{t+1}^\top L_{t+1} + \alpha I)^{-1/2}\|_F^2 \geq (1 - \delta_{2r})f(L_{t+1}, R_t). \tag{9}$$

*for all $t$ such that $\|L_t R_t^\top - X_\star\|_F \geq \rho\sigma_{r_\star}(X_\star)$*

**Remark 4.1.** *(**Intuition of initialization**) Suppose that we take infinitesimal initialization, i.e., $c_1 \to 0$, then we have $\sigma_{min}^2(R_t) \to 0$. If we take $\alpha \leq \sigma_{min}^2(R_t)$, i.e., $\alpha \to 0$, this would lead to the singularization of $(R_t^\top R_t + \alpha I)$. And if we take $\alpha \geq \sigma_{min}^2(R_t)$, this would slow down the convergence. In contrast to near-zero initialization, we emphasize that the initialization scale $c_1$ must have a lower bound $c_{init}$. This lower bound ensures that, in the first stage, APGD can converge stably and rapidly to a point that is very close to the ground truth.*

However, as $L_t R_t^\top$ converges to $X_\star$, $\min(\sigma_{\min}(R_t), \sigma_{\min}(L_t))$ converges to 0, leading to the singularization of $R_t^\top R_t$. Therefore, inequality (8) would be loose for bounding the preconditoned gradient. When $\|L_t R_t^\top - X_\star\|_F \leq \rho\sigma_{r_\star}(X_\star)$, we enter the stage 2.

### 4.2.2 STAGE 2

In this stage, since $\min(\sigma_{\min}(R_t), \sigma_{\min}(L_t))$ is relatively small, inequality (8) would be impossible to lower bound the preconditioned gradient. As a result, a new approach has been adopted in this phase. First, we introduce a lemma to lower bound the original gradient.

**Lemma 4.** *Suppose that the linear map $\mathcal{A}$ satisfies the RIP with parameters $(2r, \delta_{2r})$, and $\|L_t R_t^\top - X_\star\|_F \leq \rho\sigma_{r_\star}(X_\star)$ with $0 < \rho < 1$, then we have*

$$\|\nabla_L f(L_t, R_t)\|_F \geq \|L_t R_t^\top - X_\star\|_F \|Y_1^\star R_t^\top\|_F (\cos\theta - \delta_{2r}) \tag{10}$$

*where*

$$\cos\theta = \frac{\langle L_t R_t^\top - X_\star, Y_1 R_t^\top \rangle}{\|L_t R_t^\top - X_\star\|_F \|Y_1 R_t^\top\|_F} \geq \sqrt{1 - \rho^2} \tag{11}$$

*and $Y_1^\star$ is a corresponding maximizer for (10) satisfies $\|Y_1^\star\|_F = 1$. And we also have*

$$\|\nabla_L f(L_{t+1}, R_t)\|_F \geq \|L_{t+1} R_t^\top - X_\star\|_F \|L_{t+1} Y_2^{\star\top}\|_F (\cos\beta - \delta_{2r}) \tag{12}$$

*where*

$$\cos\beta = \frac{\langle L_{t+1} R_t^\top - X_\star, L_{t+1} Y_2^\top \rangle}{\|L_{t+1} R_t^\top - X_\star\|_F \|L_{t+1} Y_2^\top\|_F} \geq \sqrt{1 - \rho^2} \tag{13}$$

*and $Y_2^\star$ is a corresponding maximizer for (12) satisfies $\|Y_2^\star\|_F = 1$.*

**Remark 4.2.** *This lemma shows that $\|\nabla_L f(L_t, R_t)\|_F$ is highly related to $\cos\theta$ and $\|Y_1 R_t^\top\|_F$, while $\cos\theta$ captures the alignment between the row space of $L_t R_t^\top - X_\star$ and $R_t^\top$. As $L_t R_t^\top$ is close to $X_\star$, we have $\cos\theta \geq \sqrt{1 - \rho^2}$, which indicates the error matrix $L_t R_t^\top - X_\star$ is well aligned with the row space of $R_t$. However, $R_t$ can be ill-conditioned since $R_t$ is over-parameterized, error matrix $L_t R_t^\top - X_\star$ is well aligned with an ill-conditioned space, leading to $\|Y_1^\star R_t^\top\|_F \geq \sigma_r(R_t)\|Y_1^\star\|_F$. Obviously, $\sigma_r(R_t)$ can cause the sub-linear convergence. So what we have to do is find a well-conditioned subspace of the row space of $R_t$. (Here we have only analyzed $R_t$, in fact it is similar to $L_t$.)*

Since $L_t$, $R_t$ contain ill-conditioned singular values, a straightforward idea is to exclude the effect of these ill-conditioned singular values so that $L_t R_t - X_\star$ can align towards the well-conditioned directions. Assume that $L_t$ contains $k$ large and well conditioned singular values and $r - k$ poor-conditioned singular values (near zero). Let $L = U^L S^L V^{L\top}$, $R = U^R S^R V^{R\top}$ be the SVD of $L$, $R$, and we denote

$$L_k = U_k^L S_k^L V_k^{L\top}, \quad R_k = U_k^R S_k^R V_k^{R\top}.$$

Then we rewrite inequalities (10) and (12) as

$$\|\nabla_L f(L_t, R_t)\|_F \geq \|L_t R_t^\top - X_\star\|_F \|Y_1^\star R_{t_k}^\top\|_F (\cos\theta_k - \delta_{2r}) \tag{14}$$

where

$$\cos \theta_k = \frac{\langle L_t R_t^\top - X_\star, Y_1 R_{t_k}^\top \rangle}{\|L_t R_t^\top - X_\star\|_F \|Y_1 R_{t_k}^\top\|_F} \tag{15}$$

and

$$\|\nabla_R f(L_{t+1}, R_t)\|_F \geq \|L_{t+1} R_t^\top - X_\star\|_F \|L_{t+1_k} Y_2^{\star\top}\|_F (\cos \beta_k - \delta_{2r}) \tag{16}$$

where

$$\cos \beta_k = \frac{\langle L_{t+1} R_t^\top - X_\star, L_{t+1_k} Y_2^\top \rangle}{\|L_{t+1} R_t^\top - X_\star\|_F \|L_{t+1_k} Y_2^\top\|_F}. \tag{17}$$

Motivated by (Zhang et al., 2021; 2023), we introduced two local norms and corresponding dual norms

$$\|A\|_{R,\alpha} \stackrel{\text{def}}{=} \|A P_{R,\alpha}^{1/2}\|_F, \ \|A\|_{R,\alpha}^* \stackrel{\text{def}}{=} \|A P_{R,\alpha}^{-1/2}\|_F, \ P_{R,\alpha} \stackrel{\text{def}}{=} R^\top R + \alpha I$$
$$\|A\|_{L,\alpha} \stackrel{\text{def}}{=} \|A P_{L,\alpha}^{1/2}\|_F, \ \|A\|_{L,\alpha}^* \stackrel{\text{def}}{=} \|A P_{L,\alpha}^{-1/2}\|_F, \ P_{L,\alpha} \stackrel{\text{def}}{=} L^\top L + \alpha I. \tag{18}$$

With these two local norms, we can give a lemma based on equations (15-18), bounding the preconditioned gradient with well-conditioned direction.

**Lemma 5.** *Suppose that the linear map $\mathcal{A}$ satisfies the RIP with parameters $(2r, \delta_{2r})$, then we have*

$$\|\nabla_L f(L_t, R_t)\|_{R,\alpha}^* \geq \max_{k \in \{1,2,\ldots,r\}} \frac{(\cos \theta_k - \delta_{2r})}{\sqrt{1 + \alpha/\sigma_k^2(R_t)}} \|L_t R_t^\top - X_\star\|_F,$$

*where*

$$\cos \theta_k = \frac{\langle L_t R_t^\top - X_\star, Y_1 R_{t_k}^\top \rangle}{\|L_t R_t^\top - X_\star\|_F \|Y_1 R_{t_k}^\top\|_F}, \ R_t = U^R S^R V^{R\top}, \ R_{t_k} = U_k^R S_k^R V_k^{R\top}, \tag{19}$$

*and*

$$\|\nabla_R f(L_{t+1}, R_t)\|_{L,\alpha}^* \geq \max_{k \in \{1,2,\ldots,r\}} \frac{(\cos \beta_k - \delta_{2r})}{\sqrt{1 + \alpha/\sigma_k^2(L_{t+1})}} \|L_{t+1} R_t^\top - X_\star\|_F,$$

*where*

$$\cos \beta_k = \frac{\langle L_{t+1} R_t^\top - X_\star, L_{t+1} Y_2^\top \rangle}{\|L_{t+1} R_t^\top - X_\star\|_F \|L_{t+1} Y_2^\top\|_F}, \ L_{t+1} = U^L S^L V^{L\top}, \ L_{t+1_k} = U_k^L S_k^L V_k^{L\top}, \tag{20}$$

**Remark 4.3.** *From Lemma 5, it is evident that as long as $\cos \theta_k$, $\cos \beta_k$ is sufficiently large and an appropriate damping parameter $\alpha$ is chosen, a stable lower bound for the preconditioned gradient can be achieved, ensuring linear convergence. In the following, we will demonstrate that when the distance between $L_t R_t^\top$ and $X_\star$ is sufficiently small, $\cos \theta_k$ and $\cos \beta_k$ will be large, indicating that they are well-aligned.*

**Lemma 6.** *Suppose that the linear map $\mathcal{A}$ satisfies the RIP with parameters $(2r, \delta_{2r})$ with $\delta_{2r} \leq \sqrt{2} - 1$, and $\|L_t R_t^\top - X_\star\|_F \leq \rho \sigma_{r_\star}(X_\star)$, $\rho = \frac{(1-\delta_{2r})}{2}$, then we have*

$$\|\nabla_L f(L_t, R_t)\|_{R,\alpha}^* \geq \frac{1 - \delta_{2r}}{2\sqrt{2}} \left(1 + c_4 \frac{\alpha}{\|L_t R_t^\top - X_\star\|_F^2}\right)^{-1/2} \|L_t R_t^\top - X_\star\|_F,$$

$$\|\nabla_R f(L_{t+1}, R_t)\|_{L,\alpha}^* \geq \frac{1 - \delta_{2r}}{2\sqrt{2}} \left(1 + c_5 \frac{\alpha}{\|L_{t+1} R_t^\top - X_\star\|_F^2}\right)^{-1/2} \|L_{t+1} R_t^\top - X_\star\|_F,$$

*where $c_4$ and $c_5$ are constants.*

**Remark 4.4.** *From Lemma 6, it can be seen that by selecting an proper damping parameter $\alpha$, we can obtain a $\mu_p$ that depends only on RIP constant $\delta_{2r}$. Combining this with the first stage, it becomes clear that APGD, starting from the initial point, can consistently converge at a linear rate.*

# 5 EXPERIMENTS

In this section, we conduct simulation experiments to empirically validate our theoretical results. Our experiments demonstrate that with random initialization, APGD exhibits condition number-independent linear convergence rates in matrix factorization and matrix sensing problems, even in over-parameterized settings. We compared our method with the vanilla gradient descent, alternating gradient descent, ScaledGD($\lambda$) and PrecGD, and showed that our approach achieves the fastest convergence rate. Furthermore, compared to ScaledGD($\lambda$), our method is more robust to the choice of preconditioner damping parameters $\alpha$ and step size $\eta$.

**Introduction of comparison methods** Firstly, we provide a brief introduction to the comparison methods. The methods under comparison include three FGD based methods: ScaledGD($\lambda$) (Xu et al., 2023), PrecGD (Zhang et al., 2021; 2024), and vanilla GD, along with a variant of FGD method, namely alternating gradient descent. ScaledGD($\lambda$), is a preconditioning method that achieves linear convergence from very small random initializations using a constant, very small damping parameter. PrecGD is another preconditioning algorithm that starts from spectral initialization and then leverages an exponentially decaying damping parameter to effectively recover $X_\star$ from noisy observations. Alternating GD is a modified version of vanilla GD that alternately updates the two factor matrices.

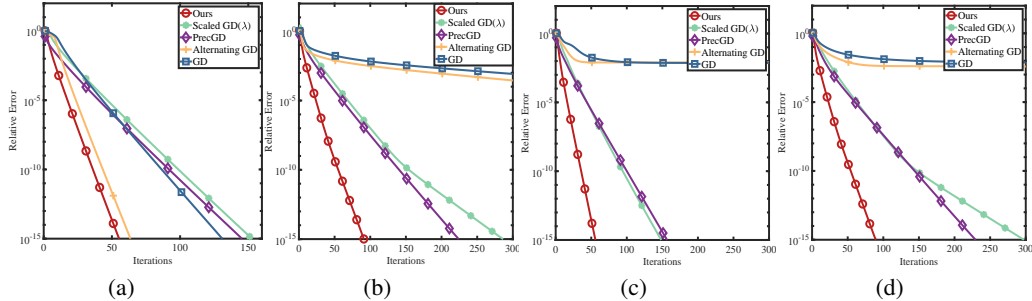

(a)          (b)          (c)          (d)

Figure 1: Comparison of the four methods in four different cases with $n_1 = n_2 = 20$, $r_\star = 5$, $m = 10n_1 r_\star$. And the step size of APGD, Alternating GD is 0.9, and the step size of ScaledGD($\lambda$) is set to be 0.6 for the best results. (a) exact rank ($r_\star = r$) and well condition ($\kappa(X_\star) = 1$) case. (b) over rank ($r = 2r_\star$) and well condition ($\kappa(X_\star) = 1$) case. (c) exact rank ($r_\star = r$) and ill-condition ($\kappa(X_\star) = 100$) case. (d) over rank ($r = 2r_\star$) and ill-condition ($\kappa(X_\star) = 100$) case.

**Experimental setup** The entries of the sensing matrix $A_i$ are sampled i.i.d with distribution $\mathcal{N}(0, \frac{1}{m})$. And the target rank-$r_\star$ matrix $X_\star \in \mathbb{R}^{n_1 \times n_2}$ with condition number $\kappa$ is generated by $U_\star \Sigma V_\star^\top$, where $U_\star$ and $V_\star$ are both orthogonal matrix and $\Sigma$ is a diagonal matrix with condition number $\kappa$.

**Comparison with several existing methods** We compared the performance of these five methods under four different scenarios with vanilla GD serves as the baseline, as shown in Figure 1. For PrecGD, we utilized the spectral initialization as described in the original work, while for the remaining four methods, we used random initialization with $c_1 = 0.1$. From Figure 1, we can draw the following conclusions:

- From Figure 1(a), we can observe that under well-conditioned and exact rank settings, the convergence rates of APGD and alternating GD are similar, while ScaledGD($\lambda$), PrecGD, and GD exhibit comparable convergence rates. Alternating methods converges significantly faster than non-alternating methods, highlighting the advantages of alternating methods.

- Over-parameterization and ill-conditioning have a pronounced impact on both alternating GD and GD, whereas their effects on ScaledGD($\lambda$), PrecGD, and APGD are less significant. Among these, APGD demonstrates a noticeably faster convergence rate than ScaledGD($\lambda$).

- The effect of over-parameterization on ScaledGD($\lambda$), PrecGD, and APGD is greater than that of ill-conditioning, which aligns with theoretical results.

**Verify the initialization scale** We investigated the impact of initialization scale on APGD and ScaledGD($\lambda$). As shown in Figure 2, when the initialization scale is sufficiently small, ScaledGD($\lambda$) first exhibits divergence, with the extent of divergence increasing as the scale decreases, thereby slowing down the convergence rate. This phenomenon has also been observed by Zhang et al. (2024). For APGD, the convergence is similarly affected by the initialization scale, where a smaller $c_1$ leads to slower convergence.

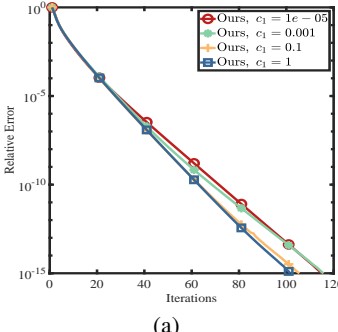 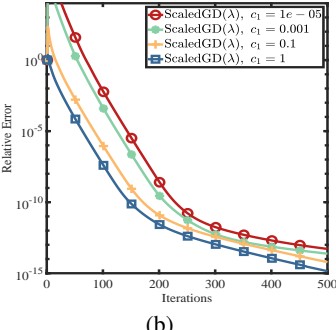

(a) (b)

Figure 2: Validating the effect of initialization scale on APGD and ScaledGD($\lambda$). We set $n_1 = n_2 = 20$, $r_\star = 5$, $k = 10$, $\kappa = 100$, $m = 10n_1r_\star$.

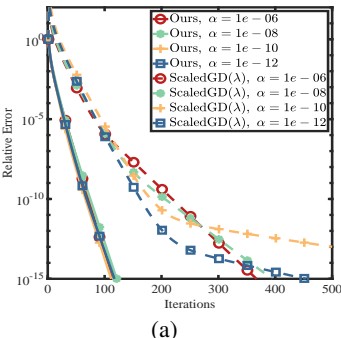 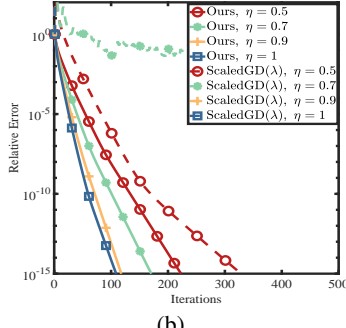

(a) (b)

Figure 3: Comparison of the sensitivity of APGD and ScaledGD($\lambda$) to different parameters (damping parameter $\alpha$, step size $\eta$). We set $n_1 = n_2 = 20$, $r_\star = 5$, $k = 10$, $\kappa = 100$, $m = 10n_1r_\star$.

**Verify the robustness of the choice of hyper-parameters** Here, we evaluate the sensitivity of APGD and ScaledGD($\lambda$) to the damping parameter and step size. As shown in Figure 3, APGD demonstrates strong robustness to both parameters, while ScaledGD($\lambda$) is more sensitive to them. Additionally, APGD allows for larger step sizes, enabling faster convergence.

## 6 CONCLUSION

We propose the APGD algorithm, to solve the low-rank asymmetric matrix sensing problem in the over-parameterized setting, where the true rank is unknown and overestimated. We theoretically and empirically demonstrate that APGD exhibits a faster convergence rate compared to previous FGD algorithms and preconditioning methods, and also possesses of great robustness to parameter sensitivity. Specifically, We develop a new two-stage analytical framework to investigate the global convergence behavior of APGD, proving that it can converge to the global optimum from the universal random initialization at a linear rate. We believe the developed framework can be naturally extended to analyze other related problems, such as 1-bit matrix sensing and low-rank matrix completion, which will be part of our future work.

**Reproducibility Statement** First, all the lemmas and theorems in the main text are provided with corresponding proofs in the appendix. Additionally, for the experiments mentioned in the paper, we have included the relevant code in the supplementary materials.

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

**The appendix contains five chapters on the additional experiments, the proof of initialization, the proof of the first stage, the proof of the second stage, and the proof of Theorem 2.**

## A  ADDITIONAL EXPERIMENTS

### A.1  EXPERIMENTS IN THE NOISY SETTING

In this section, we present additional experiments in the noisy settings. Although our theoretical analysis only covers the noiseless case, APGD also converges quickly in the presence of Gaussian noise. We compared our method with two others, including ScaledGD($\lambda$) and a preconditioned method PrecGD Zhang et al. (2024) via spectral initialization. As shown in Figure 3, APGD achieves the fastest convergence in both the exact rank and over rank settings, while achieving the same recovery error as the other two methods. Additionally, PrecGD using spectral initialization converges faster than the randomly initialized ScaledGD($\lambda$).

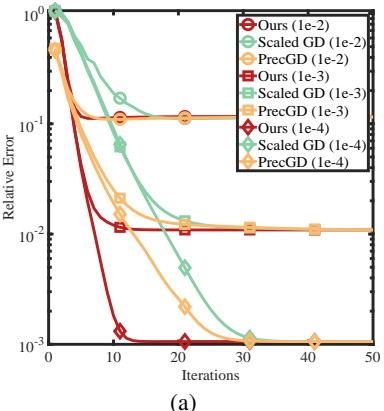
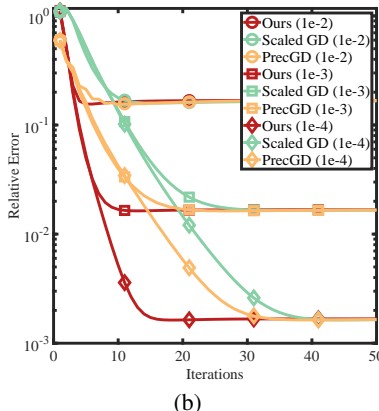

(a)                                            (b)

Figure 4: Comparison of the three methods: APGD, ScaledGD($\lambda$) and PrecGD in the noisy setting with gaussian noise of different variance: 1e-2, 1e-3, 1e-4. APGD and ScaledGD($\lambda$) use the random initialization, while PrecGD use the spectral initialization. We set $n_1 = n_2 = 20$, $r_\star = 5$, $m = 10n_1r_\star$, $\kappa = 100$. The step size of APGD is 1, while step size of other two methods is 0.5. (a): exact rank case; (b): over rank case with $r = 2r_\star = 10$.

The PrecGD we refer to is the preconditioned gradient descent method proposed by Zhang et al. (2024), which is based on spectral initialization and utilizes an exponentially decaying damping parameter. The iterative process of their algorithm is as follows:

$$X_{t+1} = X_t - \eta \nabla f(X_t)(X_t^\top X_t + \beta_t I)^{-1}, \ \beta_t = \beta_0 \beta^{t-1}.$$

Our primary focus is on comparing the sensitivity of APGD and PrecGD to the damping parameter under noisy conditions. As shown in Figure 4, both APGD and PrecGD demonstrate robustness to the damping parameter in the presence of noise. However, APGD exhibits a significantly faster convergence rate than PrecGD, despite PrecGD using spectral initialization to obtain a closer initial point.

### A.2  EXPERIMENTS WITH REGULARIZATION

As done in the work of (Tu et al., 2016), adding a regularization term is a method for solving the asymmetric matrix recovery problem. Here, we validate the feasibility of combining preconditioning with a regularization term. The objective function with the added regularization term becomes

$$\arg\min_{L,R} f(L, R) = \frac{1}{2}\|\mathcal{A}(LR^\top) - y\|_2^2 + \lambda\|L^\top L - R^\top R\|_F^2,$$

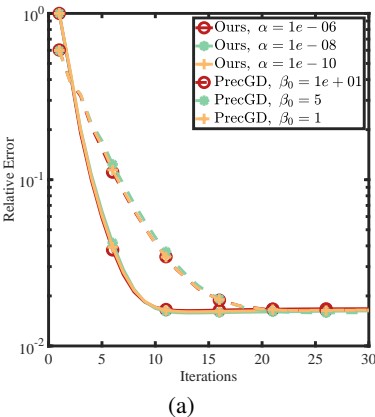 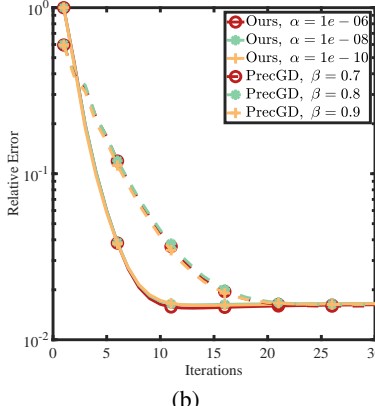

(a) (b)

Figure 5: Comparison of the sensitivity of APGD and PrecGD to damping parameter. APGD use the random initialization, while PrecGD use the spectral initialization. We set $n_1 = n_2 = 20$, $r_\star = 5$, $m = 10n_1r_\star$, $\kappa = 100$, $r = 2r_\star = 10$. The step size of APGD is 1, while step size of PrecGD is 0.5. (a): Examine the sensitivity of PrecGD to the damping parameter $\beta_0$, while also verifying the sensitivity of APGD to the damping parameter $\alpha$; (b): Examine the sensitivity of PrecGD to the damping parameter $\beta$, while also verifying the sensitivity of APGD to the damping parameter $\alpha$;.

where $\lambda$ is the regularization coefficient. Accordingly, we have the improved APGD algorithm

$$L_{t+1} = L_t - \eta \nabla_L f(L_t, R_t) \cdot (R_t^\top R_t + \alpha I)^{-1} - \eta\lambda L_t (L_t^\top L_t - R_t^\top R_t)$$
$$R_{t+1} = R_t - \eta \nabla_R f(L_{t+1}, R_t) \cdot (L_{t+1}^\top L_{t+1} + \alpha I)^{-1} - \eta\lambda R_t (R_t^\top R_t - L_{t+1}^\top L_{t+1}) \text{ (for APGD)}$$

(21)

and the ScaledGD($\lambda$) algorithm

$$L_{t+1} = L_t - \eta \nabla_L f(L_t, R_t) \cdot (R_t^\top R_t + \alpha I)^{-1} - \eta\lambda L_t (L_t^\top L_t - R_t^\top R_t)$$
$$R_{t+1} = R_t - \eta \nabla_R f(L_t, R_t) \cdot (L_t^\top L_t + \alpha I)^{-1} - \eta\lambda R_t (R_t^\top R_t - L_t^\top L_t) \text{ (for ScaledGD}(\lambda))$$

(22)

We conducted experiments comparing the performance of the original algorithms with the ones that include the regularization term. The experimental results, shown in Figure 6, indicate that adding the regularization term helps accelerate convergence for both APGD and ScaledGD($\lambda$).

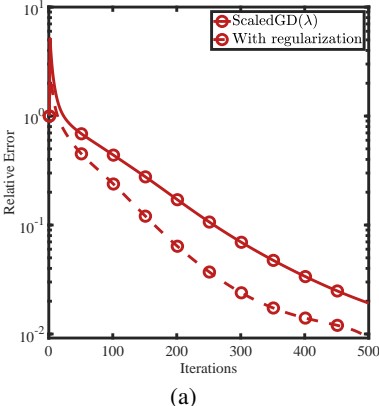 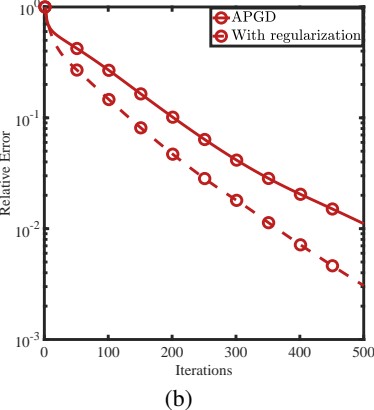

(a) (b)

Figure 6: Evaluate the effect of the regularization term on APGD and ScaledGD($\lambda$). APGD and ScaledGD($\lambda$) use the random initialization with $c_1 = 0.1$; We set $n_1 = n_2 = 20$, $r_\star = r = 5$, $m = 3n_1r_\star$, $\kappa = 100$. The step size of all algorithms are 0.1. The value of the regularization is 1.

## B  COMPARISON WITH OTHER WORKS

**Comparison with Xu et al. (2023)** The work most closely related to ours is Xu et al. (2023), which proposed a ScaledGD($\lambda$) algorithm that can converge to the global minimum from a sufficiently small initialization at a linear rate. Our approach differs from theirs in several ways. First, they focus on the recovery of symmetric positive semidefinite matrices, which is not practical, whereas we focus on the recovery of arbitrary matrices. Second, while they employ a preconditioned gradient method, we use an alternating preconditioned gradient method, which is more robust to step sizes and converges faster. Third, their method relies on an extremely small initialization, adding an extra term $\log \kappa \cdot \log(\kappa n)$ to the iteration complexity, whereas our method does not require such small initial values, thereby reducing the number of iterations needed.

**Comparision with Zhang et al. (2021; 2023)** In the second stage, our analytical approach is similar to that of Zhang et al. (2021; 2023), but have significant differences. One major difference is that Zhang et al. (2021; 2023) used a preconditioned gradient descent method, and their analysis relies on a good choice of damping parameter $\alpha$. Specifically, the convergence of preconditioned gradient descent is $(1 - \frac{\mu}{l})$. Thus, in their analysis, $\mu$ should be lower bounded, while $l$ needs to be upper bounded and these two bounds are highly related to damping parameter $\alpha$, leads to the upper bound and lower bound of $\alpha$ in each iteration, i.e. $C_{lb}\|L_t R_t^\top - X_\star\|_F \leq \alpha_t \leq C_{ub}\|L_t R_t^\top - X_\star\|_F$.

However, in our analysis only the lower bound of $\mu$ is related to $\alpha$, and the upper bound of $l$ is only related to RIP constant $\delta_{2r}$. And this is what the alternating gradient method gives us. Therefore, we can take $\alpha \leq \mathcal{O}(\|L_T R_T^\top - X_\star\|_F^2)$ for all $t \leq T$ iterations.

## C  PROOF OF INITIALIZATION

**Lemma 7.** *Let $A$ be an $n_1 \times n_2$ matrix with i.i.d Gaussian entries with distribution $N(0, 1)$. Then there exists an universal constant $C$ such that*

$$\|A\| \leq 3\sqrt{n_1 + n_2}$$

*with probability at least $1 - 2\exp -\frac{n_1 + n_2}{2}$.*

**Lemma 8.** *Suppose that we sample $\widetilde{L_0} \in \mathbb{R}^{n_1 \times r}$, $\widetilde{R_0} \in \mathbb{R}^{n_2 \times r}$ with i.i.d. $N(0, \sigma_1(X_\star))$ entries. For any fixed $c_1 > 0$, if we take $L_0 = \frac{c_1}{3\sqrt{n_1 + r}} \widetilde{L_0}$ and $R_0 = \frac{c_1}{3\sqrt{n_2 + r}} \widetilde{R_0}$, then with probability at least $1 - 2e^{\frac{n_1 + r}{2}} - 2e^{\frac{n_2 + r}{2}}$, we have*

$$\sigma_1(L_0) \leq c_1 \sqrt{\sigma_1(X_\star)}, \sigma_1(R_0) \leq c_1 \sqrt{\sigma_1(X_\star)}$$

$$\frac{1}{2}\|L_0 R_0^\top - X_\star\|_F^2 \leq C_2 \sigma_1^2(X_\star), C_2 = \frac{1}{2}rc_1^4 + \frac{1}{2}r_\star + \sqrt{rr_\star}c_1^2.$$

*Proof.* By Lemma 7, we have

$$\sigma_1(L_0) \leq c_1 \sqrt{\sigma_1(X_\star)} \text{ and } \sigma_1(R_0) \leq c_1 \sqrt{\sigma_1(X_\star)}$$

holds with probability at least $(1 - 2e^{\frac{n_1 + r}{2}} - 2e^{\frac{n_2 + r}{2}})$.

Then we have

$$\begin{aligned}
\|L_0 R_0^\top - X_\star\|_F^2 &\leq \frac{1}{2}\left(\|L_0 R_0^\top\|_F^2 + \|X_\star\|_F^2 + 2\|L_0 R_0^\top\|_F \|X_\star\|_F\right) \\
&\leq \left(r\|L_0 R_0^\top\|^2 + r_\star\|X_\star\|^2 + 2\sqrt{rr_\star}\|L_0 R_0^\top\|\|X_\star\|\right) \\
&\leq \left(rc_1^4 \sigma_1^2(X_\star) + r_\star \sigma_1^2(X_\star) + 2\sqrt{r_\star k}c_1^2 \sigma_1^2(X_\star)\right) \\
&= (\sqrt{rc_1^2} + \sqrt{r_\star})^2 \sigma_1^2(X_\star).
\end{aligned} \quad (23)$$

And we have

$$\sigma_k(L_0) \geq \frac{c_1 c_{\rho_1} \sqrt{\sigma_1}}{6\sqrt{n_1(n_1 + r)}}, \; \sigma_k(R_0) \geq \frac{c_1 c_{\rho_2} \sqrt{\sigma_1}}{6\sqrt{n_2(2_1 + r)}}$$

holds with probability at least $p_\rho = (1 - (Cc_{\rho_1})^{n_1 - r + 1} - e^{-n_1/c})(1 - (Cc_{\rho_2})^{n_2 - r + 1} - e^{-n_2/c})$. $\qquad\square$

# D  PROOF OF STAGE 1

In begin with stage 1, we first analyze the relationship between $f(L_{t+1}, R_{t+1})$ and $f(L_t, R_t)$.

### D.0.1  PROOF OF LEMMA 2

**Lemma 9.** *(Rewrite Lemma 2) Suppose that the linear map $\mathcal{A}(\cdot)$ satisfy the $\delta_{2r}$-RIP, considering the APGD in (48), we have*

$$f(L_{t+1}, R_t) \le f(L_t, R_t) - \eta(1 - \frac{\eta}{2}(1 + \delta_{2r}))\|\nabla_L f(L_t, R_t)(R_t^\top R_t + \alpha I)^{-1/2}\|_F^2$$

$$f(L_{t+1}, R_{t+1}) \le f(L_{t+1}, R_t) - \eta(1 - \frac{\eta}{2}(1 + \delta_{2r}))\|\nabla_R f(L_{t+1}, R_t)(L_{t+1}^\top L_{t+1} + \alpha I)^{-1/2}\|_F^2$$

$$\tag{24}$$

*Proof.*

$$f(L_{t+1}, R_t) = \frac{1}{2}\|\mathcal{A}(L_t R_t^\top - X_\star)\|_2^2$$

$$= \frac{1}{2}\left\|\mathcal{A}\left(\left[L_t - \eta\nabla_L f(L_t, R_t)(R_t^\top R_t + \alpha I)^{-1}\right]R_t^\top - X_\star\right)\right\|_2^2$$

$$= \underbrace{\frac{1}{2}\|\mathcal{A}(L_t R_t^\top - X_\star)\|_2^2}_{(a)} + \underbrace{\frac{\eta^2}{2}\left\|\mathcal{A}\left(\nabla_L f(L_t, R_t)(R_t^\top R_t + \alpha I)^{-1}R_t^\top\right)\right\|_2^2}_{(b)} \tag{25}$$

$$\underbrace{-\eta\left\langle\mathcal{A}(L_t R_t^\top - X_\star), \mathcal{A}\left(\nabla_L f(L_t, R_t)(R_t^\top R_t + \alpha I)^{-1}R_t^\top\right)\right\rangle}_{(c)}$$

For (b), we have

$$(b) = \frac{\eta^2}{2}\left\|\mathcal{A}\left(\nabla_L f(L_t, R_t)(R_t^\top R_t + \alpha I)^{-1}R_t^\top\right)\right\|_2^2$$

$$\overset{(i)}{\le} \frac{\eta^2}{2}(1 + \delta_{2r})\|\nabla_L f(L_t, R_t)(R_t^\top R_t + \alpha I)^{-1}R_t^\top\|_F^2$$

$$\overset{(ii)}{\le} \frac{\eta^2}{2}(1 + \delta_{2r})\|\nabla_L f(L_t, R_t)(R_t^\top R_t + \alpha I)^{-1/2}\|_F^2\|(R_t^\top R_t + \alpha I)^{-1/2}R_t^\top\|_2^2 \tag{26}$$

$$\overset{(iii)}{\le} \frac{\eta^2}{2}(1 + \delta_{2r})\|\nabla_L f(L_t, R_t)(R_t^\top R_t + \alpha I)^{-1/2}\|_F^2,$$

where $(i)$ using the assumption that the linear map $\mathcal{A}(\cdot)$ satisfy the $\delta_{2r}$-RIP and $\text{rank}(\nabla_L f(L_t, R_t)(R_t^\top R_t + \alpha I)^{-1}R_t^\top) \le \text{rank}(R_t)$; $(ii)$ using the fact that $\|AB\|_F \le \|A\|_F\|B\|_2$; $(iii)$ using the fact that $\sigma_{\max}[(R_t^\top R_t + \alpha I)^{-\frac{1}{2}}R_t^\top] = \frac{\sigma_{\max}(R_t)}{\sqrt{\sigma_{\max}^2(R_t) + \alpha}} \le 1$.

For $(c)$, we have

$$(c) = -\eta\left\langle\mathcal{A}(L_t R_t^\top - X_\star), \mathcal{A}\left(\nabla_L f(L_t, R_t)(R_t^\top R_t + \alpha I)^{-1}R_t^\top\right)\right\rangle$$

$$= -\eta\langle\mathcal{A}^*\mathcal{A}(L_t R_t^\top - X_\star), \nabla_L f(L_t, R_t)(R_t^\top R_t + \alpha I)^{-1}R_t^\top\rangle$$

$$= -\eta\langle\underbrace{\mathcal{A}^*\mathcal{A}(L_t R_t^\top - X_\star)R_t}_{\nabla_L f(L_t, R_t)}, \nabla_L f(L_t, R_t)(R_t^\top R_t + \alpha I)^{-1}\rangle \tag{27}$$

$$= -\eta\|\nabla_L f(L_t, R_t)(R_t^\top R_t + \alpha I)^{-1/2}\|_F^2.$$

Combining equations (25), (26) and (27), we have

$$f(L_{t+1}, R_t) \le f(L_t, R_t) - \eta(1 - \frac{\eta}{2}(1 + \delta_{2r}))\|\nabla_L f(L_t, R_t)(R_t^\top R_t + \alpha I)^{-1/2}\|_F^2. \tag{28}$$

By a similar approach, we can prove to obtain

$$f(L_{t+1}, R_{t+1}) \le f(L_{t+1}, R_t) - \eta(1 - \frac{\eta}{2}(1 + \delta_{2r}))\|\nabla_R f(L_{t+1}, R_t)(L_{t+1}^\top L_{t+1} + \alpha I)^{-1/2}\|_F^2. \tag{29}$$

Thereby, we complete the proof of Lemma 9. □

Then, we want to lower bound $\sigma_{\min}(L_t)$, $\sigma_{\min}(R_t)$. We first start from the low-rank matrix factorization problem, and then extend the results to the low-rank matrix sensing problem with the help of RIP condition.

### D.1 PROOF OF LOW-RANK MATRIX FACTORIZATION

In order to lower bound $\sigma_{\min}(L_t)$, $\sigma_{\min}(R_t)$, we first considering the low-rank matrix factorization problem

$$\min \Phi(L, R) = \frac{1}{2}\|LR^\top - X_\star\|_F^2 \ \ \text{subject to} \ \ L \in \mathbb{R}^{n_1 \times k}, \ R \in \mathbb{R}^{n_2 \times k}, \tag{30}$$

where $\operatorname{rank}(X_\star) = r \le k$. We consider to solve this problem by Alternating Scaled Gradient Descent (APGD):

$$\begin{aligned}
L_{t+1} &= L_t - \eta \nabla_L \Phi(L_t, R_t) \cdot (R_t^\top R_t + \alpha I)^{-1} \\
R_{t+1} &= R_t - \eta \nabla_R \Phi(L_{t+1}, R_t) \cdot (L_{t+1}^\top L_{t+1} + \alpha I)^{-1}.
\end{aligned} \tag{31}$$

**Lemma 10.** *Consider the APGD in (31), then we have*

$$\Phi(L_{t+1}, R_t) \le \Phi(L_t, R_t) - \eta(1 - \frac{\eta}{2})\|\nabla_L \Phi(L_t, R_t)(R_t^\top R_t + \alpha I)^{-1/2}\|_F^2,$$

$$\Phi(L_{t+1}, R_{t+1}) \le \Phi(L_{t+1}, R_t) - \eta(1 - \frac{\eta}{2})\|\nabla_R \Phi(L_{t+1}, R_t)(L_{t+1}^\top L_{t+1} + \alpha I)^{-1/2}\|_F^2. \tag{32}$$

*Proof.* According to the APGD, we have

$$\begin{aligned}
\Phi(L_{t+1}, R_t) &= \frac{1}{2}\|X_\star - L_{t+1}R_t^\top\|_F^2 \\
&= \frac{1}{2}\|X_\star - \left[L_t - \eta \nabla_L \Phi(L_t, R_t) \cdot (R_t^\top R_t + \alpha I)^{-1}\right] R_t^\top\|_F^2 \\
&= \underbrace{\frac{1}{2}\|X_\star - L_t R_t^\top\|_F^2}_{(a)} + \underbrace{\frac{\eta^2}{2}\|\nabla_L \Phi(L_t, R_t) \cdot (R_t^\top R_t + \alpha I)^{-1} R_t^\top\|_F^2}_{(b)} \\
&\quad \underbrace{- \eta \operatorname{tr}\left\{(L_t R_t^\top - X_\star)\left[\nabla_L \Phi(L_t, R_t) \cdot (R_t^\top R_t + \alpha I)^{-1} R_t^\top\right]^\top\right\}}_{(c)}
\end{aligned} \tag{33}$$

For $(b)$, we have

$$\begin{aligned}
(b) &= \frac{\eta^2}{2}\|\nabla_L \Phi(L_t, R_t) \cdot (R_t^\top R_t + \alpha I)^{-1} R_t^\top\|_F^2 \\
&\le \frac{\eta^2}{2}\|\nabla_L \Phi(L_t, R_t) \cdot (R_t^\top R_t + \alpha I)^{-\frac{1}{2}}\|_F^2 \cdot \|(R_t^\top R_t + \alpha I)^{-\frac{1}{2}} R_t^\top\|_2^2 \\
&\overset{(i)}{\le} \frac{\eta^2}{2}\|\nabla_L \Phi(L_t, R_t) \cdot (R_t^\top R_t + \alpha I)^{-\frac{1}{2}}\|_F^2,
\end{aligned} \tag{34}$$

where $(i)$ using the fact that $\sigma_{\max}[(R_t^\top R_t + \alpha I)^{-\frac{1}{2}} R_t^\top] = \frac{\sigma_{\max}(R_t)}{\sqrt{\sigma_{\max}^2(R_t) + \alpha}} \le 1$.

For $(c)$, we have

$$\begin{aligned}
(c) &= -\eta \operatorname{tr}\left\{(L_t R_t^\top - X_\star)\left[\nabla_L \Phi(L_t, R_t) \cdot (R_t^\top R_t + \alpha I)^{-1} R_t^\top\right]^\top\right\} \\
&= -\eta \operatorname{tr}\left\{\underbrace{(L_t R_t^\top - X_\star)R_t}_{\nabla_L \Phi(L_t, R_t)} \cdot (R_t^\top R_t + \alpha I)^{-1/2}\left[\nabla_L \Phi(L_t, R_t) \cdot (R_t^\top R_t + \alpha I)^{-1/2}\right]^\top\right\} \\
&= -\eta\|\nabla_L \Phi(L_t, R_t) \cdot (R_t^\top R_t + \alpha I)^{-\frac{1}{2}}\|_F^2.
\end{aligned} \tag{35}$$

Combining equations (33), (34) and (35), we have

$$\Phi(L_{t+1}, R_t) \leq \Phi(L_t, R_t) - \left(\eta - \frac{\eta^2}{2}\right) \|\nabla_L \Phi(L_t, R_t) \cdot (R_t^\top R_t + \alpha I)^{-\frac{1}{2}}\|_F^2. \qquad (36)$$

By a similar approach, we can prove to obtain

$$\Phi(L_{t+1}, R_{t+1}) \leq \Phi(L_{t+1}, R_t) - (\eta - \frac{\eta^2}{2})\|\nabla_R \Phi(L_{t+1}, R_t)(L_{t+1}^\top L_{t+1} + \alpha I)^{-1/2}\|_F^2. \qquad (37)$$

Thereby, we complete the proof of Lemma 10. □

**Lemma 11.** *By choosing a sufficiently small* $\alpha \leq \min\{\sigma_{\min}^2(L_t), \sigma_{\min}^2(R_t)\}_{t=1}^T$, *we have*

$$\|\nabla_L \Phi(L_t, R_t)(R_t^\top R_t + \alpha I)^{-1/2}\|_F^2 \geq \Phi(L_t, R_t),$$
$$\|\nabla_R \Phi(L_{t+1}, R_t)(L_{t+1}^\top L_{t+1} + \alpha I)^{-1/2}\|_F^2 \geq \Phi(L_{t+1}, R_t). \qquad (38)$$

*Proof.* We have

$$\begin{aligned}
\|\nabla_L \Phi(L_t, R_t)(R_t^\top R_t + \alpha I)^{-1/2}\|_F^2 &= \|(L_t R_t^\top - X_\star)R_t(R_t^\top R_t + \alpha I)^{-1/2}\|_F^2 \\
&\geq \sigma_{\min}^2\left(R_t(R_t^\top R_t + \alpha I)^{-1/2}\right)\|L_t R_t^\top - X_\star\|_F^2 \\
&= \frac{2}{1 + \alpha/\sigma_{\min}^2(R_t)}\Phi(L_t, R_t) \\
&\geq \Phi(L_t, R_t),
\end{aligned} \qquad (39)$$

where the last inequality follows from the choice of $\alpha$ : $\alpha \leq \min\{\sigma_{\min}^2(L_t), \sigma_{\min}^2(R_t)\}_{t=1}^T$. By a similar approach, we can prove to obtain

$$\|\nabla_R \Phi(L_{t+1}, R_t)(L_{t+1}^\top L_{t+1} + \alpha I)^{-1/2}\|_F^2 \geq \Phi(L_{t+1}, R_t). \qquad (40)$$

Thereby, we complete the proof of Lemma 11. □

**Lemma 12.** *Assume we have Lemma 8 holds, and* $0 < \eta < 2$, *then for any* $t \leq T$, *where* $T$ *is the last iteration that* $\|L_T R_T^\top - X_\star\|_F \geq \rho\sigma_r(X_\star)$, *we have*

$$\sigma_r^2(L_{t+1}) \geq (1-\gamma)\sigma_r^2(L_t), \ \sigma_r^2(R_{t+1}) \geq (1-\gamma)\sigma_r^2(R_t)$$

*where* $0 < \gamma \leq \eta_{c_1}$ *and* $\eta_{c_1} = \eta - 0.5\eta^2$.

*Proof.* Note that we have the APGD:

$$L_{t+1} = L_t + \underbrace{\eta(X_\star - L_t R_t^\top)R_t(R_t^\top R_t + \alpha I)^{-1}}_{P_{L_t}}$$

$$R_{t+1} = R_t + \underbrace{\eta(X_\star - L_{t+1}R_t^\top)^\top L_{t+1}(L_{t+1}^\top L_{t+1} + \alpha I)^{-1}}_{P_{R_t}}.$$

Define an auxiliary matrix $M_{L_t}$ and $M_{R_t}$ as

$$\begin{aligned}
M_{L_t} &= (I + F_{L_t})^\top L_t^\top L_t(I + F_{L_t}), \ F_{L_t} = (L_t^\top L_t)^{-1}L_t^\top P_{L_t} \\
M_{R_t} &= (I + F_{R_t})^\top R_t^\top R_t(I + F_{R_t}), \ F_{R_t} = (R_t^\top R_t)^{-1}R_t^\top P_{R_t}
\end{aligned} \qquad (41)$$

Below, we provide the analysis of $L_{t+1}$. The analysis for $R_{t+1}$ is analogous to that of $L_{t+1}$ and is thus omitted for brevity. It's easy to verify that

$$L_{t+1}^\top L_{t+1} - M_{L_t} = P_{L_t}^\top (I - L_t(L_t^\top L_t)^{-1}L_t^\top)P_{L_t} \succeq 0. \qquad (42)$$

Therefore, if we want to lower bound $\lambda_k(L_{t+1}^\top L_{t+1})$, we can lower bound $\lambda_k(M_{L_t})$. In order to lower bound $\lambda_k(M_{L_t})$, we introduce an auxiliary lemma (Eisenstat & Ipsen, 1995).

**Lemma 13.** *Let $A \in \mathbb{R}^{d \times d}$ be a symmetric matrix with eigenvalues $\lambda_1 \geq \lambda_2 \geq \cdots \geq \lambda_d$. Moreover, suppose $B$ is a non-singular matrix. Let $D = B^\top A B$ with eigenvalues $\hat{\lambda}_1 \geq \hat{\lambda}_2 \geq \cdots \geq \hat{\lambda}_d$. Then we have*

$$|\lambda_i - \hat{\lambda}_i| \leq |\lambda_i| \|I - B^\top B\|, \text{ for all } i.$$

By this lemma, we have

$$\lambda_k(M_{L_t}) \geq (1 - \|F_{L_t} + F_{L_t}^\top + F_{L_t} F_{L_t}^\top\|) \lambda_k(L_t^\top L_t). \tag{43}$$

Then we need to prove that $\|F_{L_t} + F_{L_t}^\top + F_{L_t} F_{L_t}^\top\| < 1$. Firstly we assume that $\|F_{L_t}\| < 1$, then we have

$$\|F_{L_t} + F_{L_t}^\top + F_{L_t} F_{L_t}^\top\| \leq 3\|F_{L_t}\| \leq \frac{3\eta\|L_t R_t^\top - X_\star\|}{\sigma_k(L_t)\sigma_k(R_t) + \alpha\sigma_k(L_t)/\sigma_k(R_t)}.$$

We use induction to prove that $\|F_{L_t} + F_{L_t}^\top + F_{L_t} F_{L_t}^\top\| \leq \gamma < 1$. For $t = 0$, we have

$$\|F_{L_0} + F_{L_0}^\top + F_{L_0} F_{L_0}^\top\| \leq \frac{3\eta\|L_0 R_0^\top - X_\star\|}{\sigma_k(L_0)\sigma_k(R_0)} \lesssim \frac{c_3\eta(c_1^2\sigma_1(X_\star) + \sigma_1(X_\star))}{\sigma_k(L_0)\sigma_k(R_0)}. \tag{44}$$

By the initialization assumption, we have

$$\sigma_k(L_0) \geq \frac{c_1 c_{\rho 1}\sqrt{\sigma_1}}{6\sqrt{n_1(n_1 + r)}}, \ \sigma_k(R_0) \geq \frac{c_1 c_{\rho 2}\sqrt{\sigma_1}}{6\sqrt{n_2(n_2 + r)}}.$$

By taking $c_{\rho 1} = 6\sqrt{n_1(n_1 + r)}$, $c_{\rho 2} = 6\sqrt{n_2(n_2 + r)}$, we have

$$\|F_{L_0} + F_{L_0}^\top + F_{L_0} F_{L_0}^\top\| < 1,$$

if we take $\eta < \frac{c_1^2}{c_3(c_1^2 + 1)}$ and $c_3 = 1 + 1/c_1^2$. Then there exits $0 < \gamma < 1$ such that $\|F_{L_0} + F_{L_0}^\top + F_{L_0} F_{L_0}^\top\| \leq \gamma$. Then we assume that

$$\|F_{L_t} + F_{L_t}^\top + F_{L_t} F_{L_t}^\top\| \leq \frac{3\eta\|L_t R_t^\top - X_\star\|_F}{\sigma_k(L_t)\sigma_k(R_t)} \leq \gamma$$

and prove that

$$\|F_{L_{t+1}} + F_{L_{t+1}}^\top + F_{L_{t+1}} F_{L_{t+1}}^\top\| \leq \gamma.$$

For $\|F_{L_{t+1}} + F_{L_{t+1}}^\top + F_{L_{t+1}} F_{L_{t+1}}^\top\|$, we have

$$\|F_{L_{t+1}} + F_{L_{t+1}}^\top + F_{L_{t+1}} F_{L_{t+1}}^\top\| \leq \frac{3\eta\|L_{t+1} R_{t+1}^\top - X_\star\|_F}{\sigma_k(L_{t+1})\sigma_k(R_{t+1})} \leq \frac{3\eta(1 - \eta_{c_1})\|L_t R_t^\top - X_\star\|_F}{(1 - \gamma)\sigma_k(L_t)\sigma_k(R_t)} \leq \gamma, \tag{45}$$

where the last inequality we use the assumption that $\eta_{c_1} \geq \gamma$. $\square$

### D.2 PROOF OF LOW-RANK MATRIX SENSING

Based on the results obtained in the matrix factorization problem, we further consider the lower bound of $\sigma_r(R_t)$ and $\sigma_r(L_t)$ in the matrix sensing problem.

**Lemma 14.** *Assume we have the same setting as Theorem 1, and $\eta_c \geq \gamma$, then we have*

$$\sigma_r^2(L_{T1}) \geq \frac{\rho^2\sigma_{r_\star}^2(X_\star)\sigma_r^2(L_0)}{2}, \ \sigma_r^2(R_{T1}) \geq \frac{\rho^2\sigma_{r_\star}^2(X_\star)\sigma_r^2(R_0)}{2}, \tag{46}$$

*where $\|L_{T_1} R_{T_1}^\top - X_\star\|_F \geq \rho\sigma_{\min}(X_\star)$ and $\|L_{T_1+1} R_{T_1+1}^\top - X_\star\|_F < \rho\sigma_{\min}(X_\star)$.*

*Proof.* We consider the low-rank matrix sensing problem

$$\min f(L_t, R_t) = \frac{1}{2}\|\mathcal{A}(L_t R_t^\top - X_\star)\|_2^2, \tag{47}$$

where $\mathcal{A} : \mathbb{R}^{n_1 \times n_2} \to \mathbb{R}^m$ denotes the linear map. This problem ca be solved by APGD:

$$
\begin{aligned}
L_{t+1} &= L_t - \eta \mathcal{A}^* \mathcal{A}(L_t R_t^\top - X_\star) R_t \cdot (R_t^\top R_t + \alpha I)^{-1} \\
R_{t+1} &= R_t - \eta \mathcal{A}^* \mathcal{A}(L_{t+1} R_t^\top - X_\star)^\top L_t \cdot (L_{t+1}^\top L_{t+1} + \alpha I)^{-1}.
\end{aligned}
\tag{48}
$$

Note that we have the APGD:

$$
L_{t+1} = \underbrace{L_t + \eta(X_\star - L_t R_t^\top) R_t (R_t^\top R_t + \alpha I)^{-1}}_{\widetilde{L_{t+1}}} + P_{L_t}
$$

$$
P_{L_t} = (\eta \mathcal{A}^* \mathcal{A}(X_\star - L_t R_t^\top) R_t \cdot (R_t^\top R_t + \alpha I)^{-1} - \eta(X_\star - L_t R_t^\top) R_t (R_t^\top R_t + \alpha I)^{-1})
$$

$$
R_{t+1} = \underbrace{R_t + \eta(X_\star - L_{t+1} R_t^\top)^\top L_{t+1} (L_{t+1}^\top L_{t+1} + \alpha I)^{-1}}_{\widetilde{R_{t+1}}} + P_{R_t}
$$

$$
P_{R_t} = (\eta \mathcal{A}^* \mathcal{A}(X_\star - L_{t+1} R_t^\top)^\top L_{t+1} \cdot (L_{t+1}^\top L_{t+1} + \alpha I)^{-1} - \eta(X_\star - L_{t+1} R_t^\top)^\top L_{t+1} (L_{t+1}^\top L_{t+1} + \alpha I)^{-1}).
$$

Define an auxiliary matrix $M_{L_t}$ and $M_{R_t}$ as

$$
M_{L_t} = (I + F_L) \widetilde{L_{t+1}}^\top \widetilde{L_{t+1}} (I + F_{L_t})^\top, \quad F_{L_t} = P_{L_t}^\top (\widetilde{L_{t+1}}^\top \widetilde{L_{t+1}})^{-1} \widetilde{L_{t+1}}^\top.
\tag{49}
$$

It's easy to verify that

$$
L_{t+1}^\top L_{t+1} - M_{L_t} = P_{L_t}^\top (I - \widetilde{L_{t+1}} (\widetilde{L_{t+1}}^\top \widetilde{L_{t+1}})^{-1} \widetilde{L_{t+1}}^\top) P_{L_t} \succeq 0.
\tag{50}
$$

Therefore, if we want to lower bound $\lambda_r(L_{t+1}^\top L_{t+1})$, we can lower bound $\lambda_r(M_{L_t})$.

Similar to the matrix factorization case, we need to bound $\|F_{L_t} + F_{L_t} + F_{L_t} F_{L_t}^\top\|$. Note that we have

$$
\|F_{L_t} + F_{L_t} + F_{L_t} F_{L_t}^\top\| \leq 3\|F_{L_t}\| \leq \frac{3\eta \|P_{L_t}\|}{\sigma_r(\widetilde{L_{t+1}})} \leq \frac{3\eta \sqrt{\delta_{2r}} \|L_t R_t^\top - X_\star\|}{\sqrt{1-\gamma} \sigma_r(L_t) \sigma_r(R_t)}
\tag{51}
$$

due to the rank-$2r$ RIP condition with constant $\delta_{2r}$. We then use induction to prove that

$$
\|F_{L_t} + F_{L_t}^\top + F_{L_t} F_{L_t}^\top\| \leq \gamma.
$$

For $t = 0$, if we take $\delta_{2r} \leq 1 - \gamma$ and combining the result in Lemma 12, then we have

$$
\|F_{L_0} + F_{L_0}^\top + F_{L_0} F_{L_0}^\top\| \leq \gamma.
$$

Assume we have $\|F_{L_t} + F_{L_t}^\top + F_{L_t} F_{L_t}^\top\| \leq \gamma$, then we proceed to prove $\|F_{L_{t+1}} + F_{L_{t+1}}^\top + F_{L_{t+1}} F_{L_{t+1}}^\top\| \leq \gamma$. We have

$$
\|F_{L_{t+1}} + F_{L_{t+1}}^\top + F_{L_{t+1}} F_{L_{t+1}}^\top\| \leq \frac{3\eta \|P_{L_{t+1}}\|_F}{\sigma_r(\widetilde{L_{t+2}})} \leq \frac{\sqrt{\delta_{2r} \cdot \frac{1+\delta_{2r}}{1-\delta_{2r}}} (1 - \eta_c)}{(1-\gamma)^{3/2}} \cdot \frac{\|L_t R_t^\top - X_\star\|_F}{\sigma_r(L_t) \sigma_r(R_t)}.
\tag{52}
$$

Since we have $\delta_{2r} \leq \sqrt{2} - 1$, then we have $\delta_{2r}(\frac{1+\delta_{2r}}{1-\delta_{2r}}) \leq 1 - \eta_c$ and $\eta_c \geq \gamma$, then we can directly use the result of Lemma 12, i.e.,

$$
\|F_{L_t} + F_{L_t} + F_{L_t} F_{L_t}^\top\| \leq \gamma.
$$

Then we have

$$
\sigma_r(L_{t+1}^\top L_{t+1}) \geq \sigma_r(M_{L_t}) \geq (1 - \gamma) \sigma_r(\widetilde{L_{t+1}}^\top \widetilde{L_{t+1}}) \geq (1 - \gamma)^2 \sigma_r(L_t^\top L_t)
\tag{53}
$$

As for $\sigma_r(R_t)$, it can be proved in a similar way, so we omit its proof and given the result directly:

$$
\sigma_r(R_{t+1}^\top R_{t+1}) \geq (1 - \gamma)^2 \sigma_r(R_t^\top R_t)
\tag{54}
$$

Then for stage 1, we have

$$
f(L_{t+1}, L_{t+1}) \leq (1 - \eta_c)^2 f(L_t, R_t), \quad \eta_c = \eta(1 - \frac{\eta}{2}(1 + \delta_{2r}))(1 - \delta_{2r}).
\tag{55}
$$

Let $T_1$ be the last iteration that $\|L_{T_1}R_{T_1}^\top - X_\star\|_F \geq \rho\sigma_{r_\star}(X_\star)$, then $\|L_{T_1+1}R_{T_1+1}^\top - X_\star\|_F \leq \rho\sigma_{r_\star}(X_\star)$. We have

$$\|L_{T_1}R_{T_1}^\top - X_\star\|_F \leq \sqrt{1+\delta_{2r}}(1-\eta_c)^{T_1}(\sqrt{r}c_1^2 + \sqrt{r_\star})\sigma_1(X_\star) \leq \rho\sigma_{r_\star}(X_\star), \tag{56}$$

where $T_1 = \Omega(\log(\kappa r))$. Then we can establish the lower bound for $\sigma_r(L_T)$ and $\sigma_r(R_T)$.

$$\begin{aligned}
\sigma_r^2(L_{T1}) &\geq (1-\gamma)^{(2T_1)}\sigma_r^2(L_0) \geq (1-\eta_c)^{(2T_1)}\sigma_r^2(L_0) \\
&\geq \frac{\|L_{T_1}R_{T_1}^\top - X_\star\|_F^2}{1+\delta_{2r}}\sigma_r^2(L_0) \geq \frac{\rho^2\sigma_{r_\star}^2(X_\star)\sigma_r^2(L_0)}{2} \\
\sigma_r^2(R_{T1}) &\geq (1-\gamma)^{(2T_1)}\sigma_r^2(R_0) \geq (1-\eta_c)^{(2T_1)}\sigma_r^2(R_0) \\
&\geq \frac{\|L_{T_1}R_{T_1}^\top - X_\star\|_F^2}{1+\delta_{2r}}\sigma_r^2(R_0) \geq \frac{\rho^2\sigma_{r_\star}^2(X_\star)\sigma_r^2(R_0)}{2}.
\end{aligned} \tag{57}$$

Therefore, by taking $\alpha = \mathcal{O}(\epsilon)$ and $c_1 \geq c_{init} = \frac{\sqrt{2\alpha}\kappa}{\rho}$, we have

$$\alpha \leq \min\{\sigma_r^2(L_{T1}), \sigma_r^2(R_{T1})\}.$$

$\square$

### D.3 PROOF OF LEMMA 3

**Lemma 15.** *(Recall Lemma 3) Suppose that the linear map $\mathcal{A}(\cdot)$ satisfy the $\delta_{2r}$-RIP, then by choosing a infinitesimal $\alpha \leq \min\{\sigma_{\min}^2(L_t), \sigma_{\min}^2(R_t)\}_{t=1}^T$, we have*

$$\begin{aligned}
\|\nabla_L f(L_t, R_t)(R_t^\top R_t + \alpha I)^{-1/2}\|_F^2 &\geq (1-\delta_{2r})f(L_t, R_t), \\
\|\nabla_R f(L_{t+1}, R_t)(L_{t+1}^\top L_{t+1} + \alpha I)^{-1/2}\|_F^2 &\geq (1-\delta_{2r})f(L_{t+1}, R_t).
\end{aligned} \tag{58}$$

*Proof.*

$$\begin{aligned}
\|\nabla_L f(L_t, R_t)(R_t^\top R_t + \alpha I)^{-1/2}\|_F^2 &= \|\mathcal{A}^*\mathcal{A}(L_tR_t^\top - X_\star)R_t(R_t^\top R_t + \alpha I)^{-1/2}\|_F^2 \\
&\geq \sigma_{\min}^2\left(R_t(R_t^\top R_t + \alpha I)^{-1/2}\right)\|\mathcal{A}^*\mathcal{A}(L_tR_t^\top - X_\star)\|_F^2.
\end{aligned} \tag{59}$$

For $\|\mathcal{A}^*\mathcal{A}(L_tR_t^\top - X_\star)\|_F$, we have

$$\begin{aligned}
\|\mathcal{A}^*\mathcal{A}(L_tR_t^\top - X_\star)\|_F &= \max_{Y:\|Y\|_F \leq 1}\langle\mathcal{A}^*\mathcal{A}(L_tR_t^\top - X_\star), Y\rangle \\
&= \max_{Y:\|Y\|_F \leq 1}\langle\mathcal{A}(L_tR_t^\top - X_\star), \mathcal{A}(Y)\rangle \\
&\overset{(i)}{\geq} \langle\mathcal{A}(E_t), \mathcal{A}(\frac{E_t}{\|E_t\|_F})\rangle = \frac{\|\mathcal{A}(E_t)\|_2^2}{\|E_t\|_F} \\
&\overset{(ii)}{\geq} \sqrt{(1-\delta_{2r})}\|\mathcal{A}(E_t)\|_2,
\end{aligned} \tag{60}$$

where in $(i)$ we denote $L_tR_t^\top - X_\star$ as $E_t$ for convenience and construct a specific $Y = \frac{E_t}{\|E_t\|_F}$; $(ii)$ using the fact that $\|E_t\|_F \leq \frac{1}{\sqrt{1-\delta_{2r}}}\|\mathcal{A}(E_t)\|_2$.

Therefore, we have

$$\begin{aligned}
\|\nabla_L f(L_t, R_t)(R_t^\top R_t + \alpha I)^{-1/2}\|_F^2 &\geq 2\sigma_{\min}^2\left(R_t(R_t^\top R_t + \alpha I)^{-1/2}\right)(1-\delta_{2r})f(L_t, R_t) \\
&= \frac{2}{1 + \alpha/\sigma_{\min}^2(R_t)}(1-\delta_{2r})f(L_t, R_t) \\
&\overset{(i)}{\geq} (1-\delta_{2r})f(L_t, R_t),
\end{aligned} \tag{61}$$

where $(i)$ due to the choice of $\alpha$ : $\alpha \leq \min\{\sigma_{\min}^2(L_t), \sigma_{\min}^2(R_t)\}_{t=1}^T$.

By a similar approach, we can prove to obtain

$$\|\nabla_R f(L_{t+1}, R_t)(L_{t+1}^\top L_{t+1} + \alpha I)^{-1/2}\|_F^2 \geq (1-\delta_{2r})f(L_{t+1}, R_t). \tag{62}$$

Thereby, we complete the proof of Lemma 15. $\square$

# E    PROOF OF STAGE 2

## E.1    PROOF OF LEMMA 4

**Lemma 16.** *(**Rewrite Lemma 4**) Suppose that the linear map $\mathcal{A}$ satisfies the RIP with parameters* $(2r, \delta_{2r})$, *and* $\|L_t R_t^\top - X_\star\|_F \leq \rho \sigma_{r_\star}(X_\star)$ *with* $0 < \rho < 1/2$, *then we have*

$$\|\nabla_L f(L_t, R_t)\|_F \geq \|L_t R_t^\top - X_\star\|_F \|Y_1^\star R_t^\top\|_F (\cos\theta - \delta_{2r}) \tag{63}$$

*where*

$$\cos\theta = \frac{\langle L_t R_t^\top - X_\star, Y_1 R_t^\top \rangle}{\|L_t R_t^\top - X_\star\|_F \|Y_1 R_t^\top\|_F} \geq \sqrt{1 - \rho^2} \tag{64}$$

*and $Y_1^\star$ is a corresponding maximizer for (63) satisfies $\|Y_1^\star\|_F = 1$. And we also have*

$$\|\nabla_L f(L_{t+1}, R_t)\|_F \geq \|L_{t+1} R_t^\top - X_\star\|_F \|L_{t+1} Y_2^\top\|_F (\cos\beta - \delta_{2r}) \tag{65}$$

*where*

$$\cos\beta = \frac{\langle L_{t+1} R_t^\top - X_\star, L_{t+1} Y_2^\top \rangle}{\|L_{t+1} R_t^\top - X_\star\|_F \|L_{t+1} Y_2^{\star\top}\|_F} \geq \sqrt{1 - \rho^2} \tag{66}$$

*and $Y_2^\star$ is a corresponding maximizer for (63) satisfies $\|Y_2^\star\|_F = 1$.*

*Proof.* In order to prove lemma 4, we should rewrite the Frobenius norm of gradient via the definition of Frobenius norm and RIP condition, which is Lemma 17.

**Lemma 17.** *Suppose that the linear map $\mathcal{A}$ satisfies the RIP with parameters $(2r, \delta_{2r})$, then we have*

$$\|\nabla_L f(L_t, R_t)\|_F \geq \max_{\|Y_1\|_F \leq 1} \langle L_t R_t^\top - X_\star, Y_1 R_t^\top \rangle - \delta_{2r} \|L_t R_t^\top - X_\star\|_F \|Y_1 R_t^\top\|_F$$

$$\|\nabla_R f(L_{t+1}, R_t)\|_F \geq \max_{\|Y_2\|_F \leq 1} \langle L_{t+1} R_t^\top - X_\star, L_{t+1} Y_2^\top \rangle - \delta_{2r} \|L_{t+1} R_t^\top - X_\star\|_F \|L_{t+1} Y_2^\top\|_F. \tag{67}$$

*Proof.* This lemma is an extension of Lemma 14 in Zhang et al. (2021). The proof of this lemma can be obtained from the proof of Lemma 14 with simple modifications. □

With Lemma 17, we need to prove the lower bounds of $\cos\theta$ and $\cos\beta$. We start from upper bounding $\sin\theta$ and $\sin\beta$, then use the relationship between $\cos\theta$ and $\sin\theta$ to bound $\cos\theta$ and $\cos\beta$. We have the following upper bounds for $\sin\theta$ and $\sin\beta$.

**Lemma 18.** *Let $L = U^L S^L V^{L\top}$ and $R = U^R S^R V^{R\top}$ be the SVD of L and R, respectively, and $U_k^L$ and $U_k^R$ denote the matrix of first $k$ columns of $U^L$ and $U^R$ correspondingly. Suppose that $\|LR^\top - X_\star\|_F \leq \rho \sigma_{r_\star}(X_\star)$. Then for all $r \geq k \geq r_\star$, we have*

$$\frac{\|(I - U_k^L U_k^{L\top}) X_\star\|_F}{\|LR^\top - X_\star\|_F} \leq \rho, \quad \frac{\|X_\star (I - U_k^R U_k^{R\top})\|_F}{\|LR^\top - X_\star\|_F} \leq \rho. \tag{68}$$

This lemma is an extension of Lemma 6 in Cheng & Zhao (2024). The proof of this lemma can be obtained from the proof of Lemma 6 with simple modifications. Then we have

$$\sin\theta = \frac{\|(L_t R_t^\top - X_\star)(I - U^R U^{R\top})\|_F}{\|L_t R_t^\top - X_\star\|_F} \leq \rho \rightarrow \cos\theta \geq \sqrt{1 - \rho^2}.$$

$$\sin\beta = \frac{\|(I - U^L U^{L\top})(L_{t+1} R_t^\top - X_\star)\|_F}{\|L_{t+1} R_t^\top - X_\star\|_F} \leq \rho \rightarrow \cos\beta \geq \sqrt{1 - \rho^2}, \tag{69}$$

completing the proof. □

### E.2 PROOF OF LEMMA 5

As we establish the lower bound for the original gradient, using this result, we then proceed the proof of the lower bound for the gradient under the new local norms.

**Lemma 19.** *(Rewrite Lemma 5) Suppose that the linear map $\mathcal{A}$ satisfies the RIP with parameters $(2r, \delta_{2r})$, then we have*

$$\|\nabla_L f(L_t, R_t)\|_{R,\alpha}^* \geq \max_{k \in \{1,2,\dots,r\}} \frac{(\cos\theta_k - \delta_{2r})}{\sqrt{1 + \alpha/\sigma_k^2(R_t)}} \|L_t R_t^\top - X_\star\|_F,$$

*where*

$$\cos\theta_k = \frac{\langle L_t R_t^\top - X_\star, Y_1 R_{t_k}^\top \rangle}{\|L_t R_t^\top - X_\star\|_F \|Y_1 R_{t_k}^\top\|_F}, \ R_t = U^R S^R V^{R\top}, \ (R_t)_k = U_k^R S_k^R V^R{}_k^\top, \quad (70)$$

*and*

$$\|\nabla_R f(L_{t+1}, R_t)\|_{L,\alpha}^* \geq \max_{k \in \{1,2,\dots,r\}} \frac{(\cos\beta_k - \delta_{2r})}{\sqrt{1 + \alpha/\sigma_k^2(L_{t+1})}} \|L_{t+1} R_t^\top - X_\star\|_F,$$

*where*

$$\cos\beta_k = \frac{\langle L_{t+1} R_t^\top - X_\star, L_{t+1} Y_2^\top \rangle}{\|L_{t+1} R_t^\top - X_\star\|_F \|L_{t+1} Y_2^\top\|_F}, \ L_{t+1} = U^L S^L V^{L\top}, \ (L_{t+1})_k = U_k^L S_k^L V^L{}_k^\top, \quad (71)$$

*Proof.* By the definition of local norm, we have

$$\|\nabla_L f(L_t, R_t)\|_{R,\alpha}^* \geq \max_{\|Y_1\|_{R,\alpha} \leq 1} \langle L_t R_t^\top - X_\star, Y_1 R_t^\top \rangle - \delta_{2r} \|L_t R_t^\top - X_\star\|_F \|Y_1 R_t^\top\|_F,$$

$$= \|L_t R_t^\top - X_\star\|_F \|Y_1^\star(R_t^\top)_k\|_F (\cos\theta_k - \delta_{2r}).$$

For $\|Y_1^\star(R_t^\top)_k\|_F$, we have

$$\|Y_1^\star(R_t^\top)_k\|_F \geq \sigma_k(R_t)\|Y_1^\star\|_F = \sigma_k(R_t(R_t^\top R_t + \alpha I)^{-1/2})\|Y_1^\star\|_{R,\alpha} = \frac{1}{\sqrt{1 + \alpha/\sigma_k^2(R_t)}},$$

leading to

$$\|\nabla_L f(L_t, R_t)\|_{R,\alpha}^* \geq \max_{k \in \{1,2,\dots,r\}} \frac{(\cos\theta_k - \delta_{2r})}{\sqrt{1 + \alpha/\sigma_k^2(R_t)}} \|L_t R_t^\top - X_\star\|_F$$

Similarly, we have

$$\|\nabla_R f(L_{t+1}, R_t)\|_{L,\alpha}^* \geq \max_{k \in \{1,2,\dots,r\}} \frac{(\cos\beta_k - \delta_{2r})}{\sqrt{1 + \alpha/\sigma_k^2(L_{t+1})}} \|L_{t+1} R_t^\top - X_\star\|_F.$$

□

### E.3 PROOF OF LEMMA 6

**Lemma 20.** *Suppose that the linear map $\mathcal{A}$ satisfies the RIP with parameters $(2r, \delta_{2r})$, and $\|L_t R_t^\top - X_\star\|_F \leq \rho\sigma_{r_\star}(X_\star)$, then we have*

$$\frac{\|\nabla_L f(L_t, R_t)\|_{R,\alpha}}{\|L_t R_t^\top - X_\star\|_F} \geq \frac{1 - \delta_{2r}}{2}\left(1 + \alpha\frac{c_6\sigma_1(X_\star)(r - r_\star)}{(1 - \delta_{2r})^2\|L_t R_t^\top - X_\star\|_F^2}\right)^{-1/2},$$

$$\frac{\|\nabla_L f(L_{t+1}, R_t)\|_{L,\alpha}}{\|L_{t+1} R_t^\top - X_\star\|_F} \geq \frac{1 - \delta_{2r}}{2}\left(1 + \alpha\frac{c_7\sigma_1(X_\star)(r - r_\star)}{(1 - \delta_{2r})^2\|L_{t+1} R_t^\top - X_\star\|_F^2}\right)^{-1/2}.$$

*Proof.* We set $\rho = \frac{1 - \delta_{2r}}{2}$ with $\delta_{2r} \leq \sqrt{2} - 1$. We first start from the exact rank case $(r = r_\star)$. By Weyl's inequality, we have

$$\sigma_{r_\star}(L_t R_t^\top) = \sigma_{r_\star}(X_\star + L_t R_t^\top - X_\star)$$
$$\geq \sigma_{r_\star}(X_\star) - \|L_t R_t^\top - X_\star\|_F \geq (\sqrt{2/(1 - \delta_{2r})})\|L_t R_t^\top - X_\star\|_F. \quad (72)$$

And then we want to establish a relationship between $\cos\theta_k$ and $\sigma_{k+1}(L_t R_t^\top)$. Note that we have

$$\frac{\|(I - U_k^L U_k^{L\top})X_\star\|_F}{\|LR^\top - X_\star\|_F} \leq \rho, \quad \frac{\|X_\star(I - U_k^R U_k^{R\top})\|_F}{\|LR^\top - X_\star\|_F} \leq \rho \tag{73}$$

from lemma 18. Then we have

$$\sin^2\theta_k = \frac{\|(L_t R_t^\top - X_\star)(I - U_k^R U_k^{R\top})\|_F^2}{\|L_t R_t^\top - X_\star\|_F^2} \leq \rho^2 + \frac{(r-k)\sigma_{k+1}^2(L_t R_t^\top)}{\|L_t R_t^\top - X_\star\|_F^2}$$

$$\sin\beta_k^2 = \frac{\|(I - U_k^L U_k^{L\top})(L_{t+1} R_t^\top - X_\star)\|_F^2}{\|L_{t+1} R_t^\top - X_\star\|_F^2} \leq \rho^2 + \frac{(r-k)\sigma_{k+1}^2(L_{t+1} R_t^\top)}{\|L_{t+1} R_t^\top - X_\star\|_F^2}. \tag{74}$$

Then we establish the relationship as following.

**Lemma 21.** *Suppose that we have* $\|L_t R_t^\top - X_\star\|_F^2 \leq \rho^2 \sigma_{r_\star}^2(X_\star)$, $\rho = \frac{1 - \delta_{2r}}{2}$ *with* $\delta_{2r} \leq \sqrt{2} - 1$, *then we have*

$$\frac{\sigma_{k+1}(L_t R_t^\top)(r-k)}{\|L_t R_t^\top - X_\star\|_F^2} - \frac{(1-\delta_{2r})^2}{4} \geq \frac{(1+\delta_{2r})^2}{4} - \cos^2\theta_k.$$

*Proof.* This lemma is a variant version of Lemma 27 in Zhang et al. (2023). By setting $L = \delta_{2r} + 1$, and $\mu = 1 - \delta_{2r}$, the proof of this lemma can be generalized from the proof of Lemma 27 Zhang et al. (2023). □

We start from $k = r_\star$. From equation 73, we have

$$\sigma_{r_\star}(R_t) \geq \frac{1}{1 + \sqrt{2c_5\sigma_1(X_\star)}}\|L_t R_t^\top - X_\star\|_F. \tag{75}$$

If $\cos\theta_{r_\star} \geq \frac{1+\delta_{2r}}{2}$, then substituting (76) into Lemma 19, we have

$$\frac{\|\nabla_L f(L_t, R_t)\|_{R,\alpha}}{\|L_t R_t^\top - X_\star\|_F} \geq \frac{1-\delta_{2r}}{2}(1 + \alpha/\sigma_k(R_t)^2)^{-1/2} \geq \frac{1-\delta_{2r}}{2}(1 + \alpha\frac{(1+\sqrt{c_6\sigma_1(X_\star)})^2}{\|L_t R_t^\top - X_\star\|_F^2})^{-1/2}. \tag{76}$$

If $\cos\theta_{r_\star} < \frac{1+\delta_{2r}}{2}$, we can use an induction method. First we consider the base case $k = r_\star$, we have $\cos\theta_k < \frac{1+\delta_{2r}}{2}$, We can use equation (74) to bound $\sigma_{k+1}(R_t)$ as:

$$\frac{\sigma_{k+1}(L_t R_t^\top)}{\|L_t R_t^\top - X_\star\|_F} \geq \frac{1}{\sqrt{r-r_\star}}\frac{\sqrt{1-\delta_{2r}^2}}{2}. \tag{77}$$

If $\cos\theta_{k+1} \geq \frac{1+\delta_{2r}}{2}$, then substituting 77 into Lemma 6 gets

$$\frac{\|\nabla_L f(L_t, R_t)\|_{R,\alpha}}{\|L_t R_t^\top - X_\star\|_F} \geq \frac{1-\delta_{2r}}{2}(1 + \frac{\alpha}{\sigma_{k+1}(R_t^2)})^{-1/2} \geq \frac{1-\delta_{2r}}{2}\left(1 + \alpha\frac{c_6\sigma_1(X_\star)(r-r_\star)}{(1-\delta_{2r})^2\|L_t R_t^\top - X_\star\|_F^2}\right)^{-1/2}. \tag{78}$$

If $\cos\theta_{k+1} < \frac{1+\delta_{2r}}{2}$, then we repeat 77 with $k \leftarrow k + 1$ until $k = r$. When $r = k$, we have $\cos\theta_r \geq \frac{1+\delta_{2r}}{2}$, since

$$\frac{\sigma_{k+1}(L_t R_t^\top)(r-k)}{\|L_t R_t^\top - X_\star\|_F^2} = 0 \rightarrow \frac{(1+\delta)^2}{4} - \cos^2\theta_k \leq 0. \tag{79}$$

Therefore, we have two bounds (76) and (78), leading to the final bound

$$\frac{\|\nabla_L f(L_t, R_t)\|_{R,\alpha}}{\|L_t R_t^\top - X_\star\|_F} \geq \frac{1-\delta_{2r}}{2}\left(1 + \alpha\frac{c_6\sigma_1(X_\star)(r-r_\star)}{(1-\delta_{2r})^2\|L_t R_t^\top - X_\star\|_F^2}\right)^{-1/2}. \tag{80}$$

Similarly, we have

$$\frac{\|\nabla_L f(L_{t+1}, R_t)\|_{L,\alpha}}{\|L_{t+1} R_t^\top - X_\star\|_F} \geq \frac{1-\delta_{2r}}{2}\left(1 + \alpha\frac{c_7\sigma_1(X_\star)(r-r_\star)}{(1-\delta_{2r})^2\|L_{t+1} R_t^\top - X_\star\|_F^2}\right)^{-1/2}. \tag{81}$$

□

## F  PROOF OF THE MAIN THEOREM

The proof of Theorem 2 is a direct combination of results in Lemmas 2-6. Due to the assumptions of Theorem 2, we have that lemmas 2-6 hold. Therefore, we have

$$f(L_{t+1}, R_t) \leq f(L_t, R_t) - \eta(1 - \frac{\eta}{2}(1 + \delta_{2r}))\|\nabla_L f(L_t, R_t)(R_t^\top R_t + \alpha I)^{-1/2}\|_F^2$$

$$f(L_{t+1}, R_{t+1}) \leq f(L_{t+1}, R_t) - \eta(1 - \frac{\eta}{2}(1 + \delta_{2r}))\|\nabla_R f(L_{t+1}, R_t)(L_{t+1}^\top L_{t+1} + \alpha I)^{-1/2}\|_F^2$$

$$(82)$$

with

$$\|\nabla_L f(L_t, R_t)(R_t^\top R_t + \alpha I)^{-1/2}\|_F^2 \geq \mu_P f(L_t, R_t)$$

$$\|\nabla_R f(L_{t+1}, R_t)(L_{t+1}^\top L_{t+1} + \alpha I)^{-1/2}\|_F^2 \geq \mu_P f(L_{t+1}, R_t),$$

$$(83)$$

where $\mu_p = \min\{1 - \delta_{2r}, \frac{1 - \delta_{2r}}{4}\} = \frac{1 - \delta_{2r}}{4}$ since $\alpha = \mathcal{O}(\epsilon^2)$, then we can obtain linear convergence

$$f(L_{t+1}, R_{t+1}) \leq (1 - \eta_c)^2 f(L_t, R_t), \ \eta_c = \eta(1 - \frac{\eta}{2}(1 + \delta_{2r}))\mu_P. \tag{84}$$

As for the recovery error, we have

$$\|L_T R_T^\top - X_\star\|_F \leq \sqrt{\frac{1}{1 - \delta_{2r}}}\sqrt{2f(L_T, R_T)} \leq (1 - \eta_c)^T \sqrt{\frac{1}{1 - \delta_{2r}}}\sqrt{2f(L_0, R_0)}$$

$$\leq (1 - \eta_c)^T \sqrt{\frac{1 + \delta_{2r}}{1 - \delta_{2r}}}\|L_0 R_0^\top - X_\star\|_F$$

$$\leq (1 - \eta_c)^T \sqrt{\frac{1 + \delta_{2r}}{1 - \delta_{2r}}}(\sqrt{r}c_1^2 + \sqrt{r_\star})\sigma_1(X_\star)$$

$$\leq (1 - \eta_c)^T C\sqrt{r}$$

$$(85)$$

where $C = \sqrt{\frac{1 + \delta_{2r}}{1 - \delta_{2r}}}(c_1^2 + 1)$ and assume $\sigma_1(X_\star) = 1$ without loss generality. Therefore, it takes $T = \Omega(\log(r/\epsilon))$ iterations to obtain a $\epsilon$-accuracy point for APGD.

