# OpenReview forum: "Efficient Over-parameterized Matrix Sensing via Alternating Preconditioned Gradient Descent"
_ICLR.cc/2025/Conference — Submitted to ICLR 2025_

### Official Review · Reviewer_YpTR · 2024-10-31

**Soundness:** 3
**Presentation:** 4
**Contribution:** 3
**Rating:** 8
**Confidence:** 4

**Summary:**

This paper considers the problem of over-parameterized matrix sensing, where the goal is to recover a low-rank matrix $X$ from a limited number of measurements. It was previously known that over-parameterization and ill-conditioning significantly slows down methods like gradient descent. In this work the authors proposed a preconditioned version of alternating GD that starts from a random initialization and converges linearly to the optimal solution.

**Strengths:**

Overall I think this paper is well-written and makes solid contributions. The main problem and its key difficulties are introduced in a natural way. The authors also state very clearly what the prior state of art is. As a result, it is easy to compare the contributions of this work with prior work.

In terms of the technological contributions, I think it is an important improvement over prior work that the proposed algorithm APGD does not require small initialization. Previous works relied on either small initialization or spectral initialization, both of which can be very expensive (although in different ways). Here the authors show that APGD converges with just random initialization, and is able to both prove this rigorously and demonstrate this in their numerical experiments.

**Weaknesses:**

I did not find any major weaknesses in this paper. However, it is worth noting that previous work which uses small initialization (such as Stoger & Soltanolkotabi, 2021) only requires rank-$r^*$ RIP for the measurement operator, whereas this work requires rank $r$ RIP. I think this fact should be noted, as it can make a big difference when the estimated rank $r$ is very large.

**Questions:**

In the weaknesses section I mentioned that the theoretical results in work rely on rank-$r$ RIP. Is this an artifact of the theoretical analysis, or is this something that actually appears in experiments? If we only have rank-$r^*$ RIP, does the method still work?

---

> ### Author Response · Authors · 2024-11-23
> **Response to Reviewer YpTR**
>
> ## Thank you for your positive review. We will address your concern in the response below.
> In response to your concerns, we provide the following explanation: Our results depend on the over-parameterized rank $r$, which is an artifact of the theoretical analysis. In practice, we do not rely on $r$. As described in Figures 1, 2, and 3 of the main text, the number of observation equations $m = 10n r_\star$, which is independent of $r$. Therefore, even though we only have rank-$r_\star$ RIP, APGD is still able to work. It is worth noting that the core focus of this paper is not on obtaining the optimal rank-$r_\star$ RIP constant. This is a separate direction of research and can be considered as part of our future work.

---

### Official Review · Reviewer_EsWW · 2024-11-01

**Soundness:** 3
**Presentation:** 2
**Contribution:** 3
**Rating:** 6
**Confidence:** 4

**Summary:**

The paper proposes a novel Alternating Preconditioned Gradient Descent (APGD) algorithm for solving the over-parameterized low-rank matrix sensing problem. The authors address key challenges such as ill-conditioning, over-parameterization, and initialization issues in matrix sensing. The proposed method achieves linear convergence from random initialization, even in the presence of high condition numbers and rank overestimation. Theoretical results are supported by experiments that demonstrate APGD's faster convergence and robustness compared to existing methods.

**Strengths:**

This paper introduces the Alternating Preconditioned Gradient Descent (APGD) algorithm, which significantly improves matrix sensing in more practical and real-world settings by relaxing some of the strict assumptions made in prior works. The authors theoretically prove that APGD can still achieve linear convergence to a good solution under these relaxed conditions, highlighting both its theoretical contributions and practical potential. The experiments are well-designed, with appropriate and comprehensive parameter settings. The results align closely with the theoretical findings, further demonstrating that APGD achieves state-of-the-art performance compared to current FGD-based methods.

**Weaknesses:**

This paper has a few shortcomings, mainly in its presentation:

1.The abstract emphasizes APGD’s advantages over FGD, but since FGD refers to a class of methods, the paper should explicitly list which algorithms in the comparisons (e.g., ScaledGD) belong to this category, especially in the experimental section. This would help readers quickly verify the claims made in the abstract by directly comparing APGD with FGD-based methods.

2. The paper distinguishes APGD from PrecGD, which is a good approach to comparing with SOTA methods. However, several other algorithms are included in the experiments, and it should be clearly stated which one serves as the baseline. If PrecGD is a key comparison, it would be beneficial to include its performance in the noisy setting of the sensitivity analysis (e.g., damping parameter), or to explain why it was excluded from certain analyses.

3.While the importance and benefits of "alternating" in alternating preconditioned gradient descent are clear from the content for me, this aspect is under-emphasized in the abstract. A stronger focus on this in the abstract would highlight its contribution more effectively.

**Questions:**

I have two questions: one concerns the experiments, and the other might go beyond the scope of this paper as a discussion.

1. How does APGD perform in comparison to PrecGD in the sensitivity analysis of the damping parameter in the noisy setting?

2. Alternating preconditioned gradient descent is one approach to solving asymmetric matrix sensing through LR factorization. However, as noted in "Low-rank Solutions of Linear Matrix Equations via Procrustes Flow", adding regularization during the optimization process is another method for asymmetric matrix recovery. Could combining preconditioning with a similar regularization technique be a promising direction for achieving better performance?

---

> ### Author Response · Authors · 2024-11-23
> **Response to Reviewer EsWW**
>
> ## Thank you for your constructive feedback. Here, we will address your concerns one by one.
> ### **Response to weakness 1**
> Thank you for pointing this out. You are correct that FGD FGD refers to a class of methods that decompose a low-rank matrix into factor matrices and update them using gradient descent. The methods mentioned in the paper, such as ScaledGD$(\lambda)$ and PrecGD, can indeed be regarded as specific instances of FGD.  We have added a description of the comparison methods in the experimental section of the revised version, as we believe this will help readers better understand our claims.
> ### **Response to weakness 2**
> Thank you for pointing this out. The baseline for the experiments in Figure 1 is vanilla gradient descent, and we have clarified this in the revised version. Additionally, we have included PrecGD as a comparison method under noisy conditions, since its primary purpose is to handle matrix sensing problems with noise. Due to page limitations in the main text, we have included the experiments under noisy conditions in the supplementary material.
> ### **Response to weakness 3**
> Thank you for your suggestion, which is crucial for improving the quality of our paper. We have emphasized the importance of 'alternating' in the revised version.
> ### **Response to question 1**
> We have added the relevant experiments in the supplementary materials due to page limitations in the main text. The experimental results show that both APGD and PrecGD are robust to the damping parameter under noisy conditions.
> ### **Response to question 2**
> We would be delighted to discuss this issue further with you. As you mentioned, combining preconditioning with similar regularization terms is indeed feasible. We conducted preliminary experiments and included the experimental setup and results in the supplementary material. The results show that adding a regularization term can accelerate convergence for both APGD and ScaledGD$(\lambda)$. However, how to find an appropriate regularization coefficient requires further investigation. Additionally, the theoretical analysis of this approach remains a promising direction for future research.

---

> > ### Comment · Reviewer_EsWW · 2024-11-24
> >
> > Thank you for your response. Since no new significant info is revealed in the response, I would like to keep my score as is.

---

### Official Review · Reviewer_D9xn · 2024-11-01

**Soundness:** 1
**Presentation:** 2
**Contribution:** 2
**Rating:** 1
**Confidence:** 5

**Summary:**

This paper proposes an alternating preconditioned gradient descent method for the over-parameterized, ill-conditioned, asymmetric low-rank matrix sensing problem. The proposed algorithm has two stages: it begins with a specified initialization and then performs alternating preconditioned gradient descent on the over-parameterized variables $L$ and $R$. The alternating preconditioning technique is said to make the algorithm robust against over-parameterization, ill-conditioning, and hyperparameters.

**Strengths:**

The proposed algorithm makes sense for addressing the acceleration of ill-conditioning and over-parameterization.

**Weaknesses:**

1. The proposed algorithm is not explained well or properly. The writing is poor.

2. The claimed contributions and the key novel techniques for the main result are not well demonstrated and supported, such as the intuition and rationale behind the proposed initialization and the role that alternating plays in robustness and acceleration.

3. For the second stage of the algorithm, it is not surprising that the damped preconditioning works to accelerate ill-conditioned and over-parameterized low-rank matrix sensing. Although the authors list many related works, they did not point out their novelty compared to existing ones, especially (Zhang et al. 2021, Xu et al. 2023). Actually, the preconditioning trick used in this paper is nearly the same as the existing works.

4. Lastly, I want to emphasize that the main issue with this paper is that the initialization scheme is neither proper nor reasonable. Therefore, its theoretical result from the initialization stage does not make sense, or at least is not as strong as the authors claim. The random initialization in this work is to sample $L_{0}$ and $R_{0}$ with i.i.d. $N(0,\sigma_{1}(X_{\star}))$. It is not reasonable because one does not know the ground truth $X_{\star}$ when initializing. This unreasonable initialization scheme makes its proof of $||L_{0} R_{0}^{\top} - X_{\star}|| \leq \rho \sigma_{r_{\star}} (X_{\star})$ quite trivial with random matrix theory and assumptions.

A more practical random initialization scheme should be like (Stoger & Soltanolkotabi, 2021), (Lee & Stoger, 2023), and (Xu et al., 2023), which consider small random initialization with i.i.d. $N(0,\frac{1}{n})$. Their focus mainly lies in the analysis of the alignment phase and saddle avoidance phase of the initialization stage. Apart from the balanced initialization, (Xiong et al., 2023) shows that the imbalanced random initialization approach can help accelerate low-rank matrix sensing.

Therefore, the analysis of this work for random initialization is outside the real research interests.

**Questions:**

1. Is there any evidence that the initialization scheme is practical?
2. Why does the alternating scheme make sense to the robustness of the choice of hyperparameters?

---

> ### Author Response · Authors · 2024-11-23
>
> Thank you for your critical review of our paper. In response to your comments, we would like to clarify the following points:
>
>
> First, the focus of our work is not on initialization but rather on demonstrating that combining alternating updates with preconditioning achieves superior results. Specifically, APGD alternates updates between the two factor matrices, setting it apart from previous preconditioned methods. This alternating strategy decomposes the original optimization problem into two stages, thereby reducing the Lipschitz constant $L_p$ for each stage and enlarging the feasible range for the step size, i.e., $\eta < \frac{1}{L_p}$.
>
> Second, intuitively, our initialization method does rely on information of $\sigma_{\max}(X_\star)$. However, this requirement is primarily to facilitate theoretical analysis.  In practical implementations of the algorithm, it is not strictly necessary to know $\sigma_{\max}(X_\star)$.
>
> Finally, we agree with your suggestion that a more general random initialization, similar to the one used in  (Stoger & Soltanolkotabi, 2021), (Lee & Stoger, 2023), and (Xu et al., 2023), would be beneficial. This is indeed an area for improvement in our future work. We are confident that the alternating strategy will remain effective under this general initialization schemes, as evidenced by our experimental results.

---

> > ### Comment · Reviewer_D9xn · 2024-11-25
> >
> > I would like to thank the authors for their responses. However, I believe this work does not meet the standards of ICLR. Here are my reasons.
> >
> > 1. In the paper, the authors highlight random initialization as a significant contribution, and random initialization is indeed of sufficient interest in this problem. However, the conclusions drawn in the paper do not align with the authors' claims. I believe this gap cannot be overlooked.
> >
> > 3. The proposed alternating PrecGD method for over-parameterized and ill-conditioned low-rank matrix sensing appears closely related to techniques presented in prior works, particularly (Zhang 2021, Cheng 2023). The proposed “alternating” strategy does have certain contributions but is insufficient, as it has already been one of the contributions of (Jia 2023), which studied alternating ScaledGD for the low-rank matrix factorization problem.
> >
> > Therefore, I would like to keep my score as is.

---

> > > ### Author Response · Authors · 2024-11-25
> > >
> > > We would like to thank the reviewer for  response. In regard to the two points you raised, we provide the following replies:
> > >
> > > 1. Random initialization is indeed one of the points we claim. Due to theoretical limitations, the condition in our paper requires knowledge of $\sigma_{\max}$. From an algorithmic perspective, such an initialization method is easy to obtain, as the initialization scale $c_1$ can be arbitrarily chosen, as long as $c_1 > c_{init}$. This has also been verified in our experiments. From a theoretical analysis perspective, we can set the initialization as $L=c_1L_0$ with $L_0$ sampled from $\mathcal{N}(0,1)$ , and impose certain constraints on the initialization scale $c_1$ to obtain a more general initialization scheme.
> > >
> > > 2. The preconditioning technique used in our work is indeed similar to methods proposed by (Zhang,2021, Cheng,2023). However, due to our use of alternating, our approach is both theoretically and experimentally superior to theirs. Additionally, the alternating scaledGD method proposed by Jia et al. for solving matrix factorization problems does indeed demonstrate the benefits of alternation, and it serves as one of the inspirations for our work. However, Jia et al. considered the non-overparameterized matrix factorization problem, which is significantly different from the overparameterized matrix sensing problem we address. The alternating strategy makes APGD more robust to step size and damping parameters, which is one of the key contributions of our work.

---

### Official Review · Reviewer_pw3m · 2024-11-04

**Soundness:** 3
**Presentation:** 2
**Contribution:** 3
**Rating:** 5
**Confidence:** 4

**Summary:**

This paper proposes a novel optimization approach, the Alternating Preconditioned Gradient Descent (APGD) method, specifically designed to address challenges in asymmetric matrix sensing problems. The APGD method is developed to tackle the unique difficulties posed by these tasks, especially when the problem is over-parameterized and the matrices involved are ill-conditioned.

The authors rigorously prove that under an over-parameterized setting, starting from a random initialization that is not too close to zero, the APGD method can achieve a linear convergence rate Furthermore, the APGD approach is shown to be more robust to large learning rates and damping parameters, which often destabilize other optimization methods in similar settings.

**Strengths:**

This work introduces a novel approach by combining the preconditioning technique to address the asymmetric matrix sensing case effectively. Unlike Zhang (2024)'s method, in which the second update step relies solely on L_t, the proposed APGD method introduces a modification: the second step updates R_{t+1} using information from L_{t+1} within the same iteration. This strategic adjustment accelerates the convergence, allowing the algorithm to approach the solution more efficiently.

Furthermore, the APGD method offers increased flexibility in parameter selection and random initializations, which significantly enhances its robustness. This flexibility enables APGD to handle larger step sizes without compromising convergence speed or stability, making it a promising choice for practical applications where traditional methods might fail or be too restrictive.

The theoretical analysis in this work follows a two-phase framework. This approach may inspire future studies to analyze optimization tasks that start from a random initialization rather than a local starting point, potentially broadening the range of problems where APGD or similar methods can be applied. The insights from this framework could serve as a basis for analyzing other matrix recovery tasks.

**Weaknesses:**

The paper's primary theoretical framework relies on the RIP, which is central to understanding the convergence guarantees. However, the RIP condition is not fully explained or discussed. For example, the choice of requiring an RIP constant smaller than sqrt(2)-1 is mentioned but not elaborated on. A detailed discussion on the role of RIP in the APGD approach can help to understand the global convergence.

And the explanation for initialization is missing in the main context, it's hard to understand directly why close to zero initialization does not help and the relation between c_init and \sigma_1(X_*).

In the experimental section, Figures 2 and 3 are presented in a format that significantly reduces clarity. The curves in these figures are zoomed in to the extent that it is challenging to distinguish between them based on the legend. Increasing the figure size or changing legend could greatly improve readability. Additionally, a comparison with the PrecGD method is missing in Figure 1. Given the relevance of PrecGD as a related approach, its inclusion in Figure 1 would provide a more comprehensive comparison and highlight the performance differences between APGD and this established baseline.

The paper may need thorough proofreading, especially regarding notation. Some symbols and terms are borrowed from reference papers but are not defined within this paper. This lack of consistent definitions can hinder readability and force the reader to consult external sources, interrupting the flow of understanding. Furthermore, there are minor mistakes in the mathematical expressions, both in the main text and in the appendix. While these errors might not compromise the overall framework or the validity of the theoretical results, they introduce unnecessary obstacles that may lead to misunderstandings.

**Questions:**

Typos and small mistakes in math expressions I found and not complete:
Line 145 Local -> local;
Line 201 should be 0<=delta_r<1;
Line 204 the definition of RIP is defined on ||M||_F^2;
Line 207 Lemma 1 should be a high probability result, that is with probability 1- exp(-...);
Line 250 does not define sigma_min before;
Line 256 right hand side of equation should be 1/2*||A(L_{t+1}R_t^top -X_*)||_2^2;
Line 259 third line of eqn4 (a-b)^2 =a^2+b^2-2ab, the -0.5\eta should multiply by 2;
Line 265 missing );
Line 305 Lt ->L_t;
Line 309 have not defined A^*;
Line 339 reference of equation is not 64 but 62;
Line 397 should be L_t R_t^top;
Line 862 eqn36 should be (1-\eta/2);
Line 1049 second line missing coefficient 2;
Line 1070 same as Line 339;
Line 1239 in eqn 83 there should not be ^\top for (1-\eta_c)


In line 412, the authors assert that their results do not require a bound on L, the gradient Lipschitz constant, but are solely dependent on \delta_2 r. However, the RIP constant is linked with the restricted strongly convex and smooth property. Specifically, the RIP is equivalent to this property with a condition number of (1 + \delta)/(1 - \delta), suggesting that the Lipschitz constant L cannot be entirely disregarded. The authors may need to clarify this relationship and address whether it is possible to completely eliminate the dependence on L in their analysis.

---

> ### Author Response · Authors · 2024-11-23
> **Resopnse to Reviewer pw3m (1/2)**
>
> ## Thank you for your positive review and thorough feedback. Below, we would like to take this opportunity to discuss in detail the concerns and questions raised by you:
> ### **Response to weakness "The paper's primary theoretical framework relies on the RIP..."**
> Thank you for your professional suggestions. The Restricted Isometry Property (RIP) is indeed crucial for the analysis in our paper. RIP is a commonly used condition in the field of compressed sensing, which states that the operator $\mathcal{A}(\cdot)$ approximately preserves the distances between low-rank matrices. It serves as a bridge between fully observed and partially observed data. We can first analyze the population case and then extend the results to the sample case using the RIP condition. Without the RIP condition, it may be necessary to derive the results using certain concentration inequalities. Regarding the choice of the RIP constant, the value we selected ensures that $\sigma_{\min}(R_t)$ and $\sigma_{\min}(L_t)$ exhibit a linear convergence rate in the initial phase. While the constant might be optimized further, doing so is not the primary focus of our work. We have included a discussion on the RIP condition in the revised version of the manuscript.
>
> ### **Response to weakness "And the explanation for initialization is missing..."**
>
> Thank you for pointing this out. Let us first explain why close to zero initialization does not help. As is well known, gradient dominance is an important property for establishing the convergence of gradient-based methods. The larger the gradient dominance constant $\mu$, the faster the algorithm converges. Based on the analysis presented in our paper, we have
> $\mu_p=\frac{2(1-\delta_{2r})}{1+\alpha/\sigma_{\min}^2(R_t)}$.
>
> Suppose that we take infinitesimal initialization, i.e., $c_1\to 0$, then we have $\sigma_{min}^2(R_t)\to 0$. If we take $\alpha \le \sigma_{min}^2(R_t)$, i.e., $\alpha \to 0$, this would lead to the singularization of $(R_t^\top R_t+\alpha I)$ and divergence of the algorithm. And if we take $\alpha \ge \sigma_{min}^2(R_t)$, this would lead to a small $\mu_P$, then slow down the convergence. Moreover, a series of studies [1,2,3] have shown that near-zero initialization typically requires a longer time to converge.
>
> In contrast to near-zero initialization, we emphasize that the initialization scale $c_1$ must have a lower bound $c_{init}=\sqrt{\frac{2\alpha}{\sigma_1(X_\star)\sigma_{r_\star}^2(X_\star)\rho^2}}$. This lower bound ensures that, in the first stage, APGD can converge stably and rapidly to a point that is very close to the ground truth.
> ### **Response to weakness "In the experimental section, Figures 2 and 3 are presented..."**
> Thank you for your suggestion. We have added a comparison with PrecGD in Figure 1 as per your recommendation, and we have also adjusted the sizes of Figures 2 and 3.
> ### **Response to weakness 'The paper may need thorough proofreading...' and question 1 'Typos and small mistakes in math...'**
> Thank you for carefully reviewing and pointing out the typos and small mistakes in the paper. We have corrected these issues in the main text and thoroughly reviewed the entire paper to ensure that no similar errors remain.

---

> > ### Author Response · Authors · 2024-11-23
> > **Resopnse to Reviewer pw3m (2/2)**
> >
> > ### **Response to question 2 'In line 412, the authors...'**
> > Thank you for pointing this out. The RIP constant is indeed linked with the restricted strongly convex and smooth properties. Our original description was not clear, what we intend to convey is that the upper bound of $ L $ is independent of the damping parameter and depends only on the RIP constant $ \delta $. We have emphasized this in the revised version of the manuscript . Note that We have included the original comparative analysis with PrecGD in Appendix B due to space limitations.
> >
> >
> > As for the question of 'whether it is possible to completely eliminate the dependence on $L$ in the analysis', unfortunately, our answer is no. Since the Lipschitz constant $L$ is closely tied to the RIP constant, and most existing works on matrix sensing rely on the RIP condition (e.g., works [4, 5, 6]), we are currently unable to completely eliminate the dependence on $L$ in the analysis.
> >
> >
> > [1] Dominik Stöger and Mahdi Soltanolkotabi. Small random initialization is akin to spectral learning: Optimization and generalization guarantees for overparameterized low-rank matrix reconstruction. Advances in Neural Information Processing Systems,34:23831–23843,2021.
> >
> > [2] Mahdi Soltanolkotabi, Dominik Stoger, and Changzhi Xie. Implicit balancing and regularization: Generalization and convergence guarantees for overparameterized asymmetric matrix sensing. In
> > The Thirty Sixth Annual Conference on Learning Theory, pp. 5140–5142. PMLR, 2023.
> >
> > [3] Xingyu Xu, Yandi Shen, Yuejie Chi, and Cong Ma. The power of preconditioning in overparameterized low-rank matrix sensing. In International Conference on Machine Learning, pp. 38611-38654. PMLR, 2023.
> >
> > [4] Zhang G, Fattahi S, Zhang R Y. Preconditioned Gradient Descent for Overparameterized Nonconvex Burer--Monteiro Factorization with Global Optimality Certification[J]. Journal of Machine Learning Research, 2023, 24(163): 1-55.
> >
> > [5] Ma Z, Molybog I, Lavaei J, et al. Over-parametrization via lifting for low-rank matrix sensing: Conversion of spurious solutions to strict saddle,
> > points[C]//International Conference on Machine Learning. PMLR, 2023: 23373-23387.
> >
> > [6] Chen Y, Lavaei J. Measurement Manipulation of the Matrix Sensing Problem to Improve Optimization Landscape [J].

---

> ### Author Response · Authors · 2024-11-25
>
> Dear Reviewer pw3m,
>
> Please let us know if your queries have been addressed satisfactorily. As mentioned in our response, we've thoroughly incorporated your feedback, along with suggestions from the other reviewers. We hope that our response has positively influenced your perception of our work.
>
> If you require further clarifications to potentially reconsider your score, we are enthusiastic about engaging in further discussion. Please do not hesitate to contact us. We highly value the generous contribution of your time to review our paper.

---

> > ### Comment · Reviewer_pw3m · 2024-11-26
> >
> > Thank authors for their responses. However, I would like to maintain my score.
> >
> > The novelty is one major concern, as preconditioning idea and alternating GD idea are widely used in Matrix Sensing.
> >
> > And the theoretical analysis hasn't convinced me why combining these two will have advantage on choice of hyper-parameters and why the proposed method can deal with ill-conditioned case.
> >
> > Also, I don't think authors have carefully checked their updated pdf. The figure 3 is on page 11...even more than the limit for 10 pages.

---

> > > ### Author Response · Authors · 2024-11-26
> > >
> > > Thank you for your response.
> > >
> > > First, while both preconditioning and alternating have been applied in matrix sensing, the effect of combining the two approaches in matrix sensing is not yet clear. Our findings indicate that combining these strategies improves robustness with respect to hyperparameter selection.
> > >
> > > Specifically, the alternating strategy decomposes the optimization process into two steps, thereby reducing the Lipschitz constant for each step and allowing for a larger step size, i.e., $\eta < 1/L_p$. From another perspective, under the alternating strategy, the $L_p$ constant in each iteration is independent of the damping parameter. This provides a clear improvement over works such as (Zhang, 2023), where the damping parameter is related to the $L_p$ constant, requiring more careful tuning of the damping parameter.
> > >
> > > The reason why APGD can handle ill-conditioned cases is that APGD functions similarly to Newton's method.  However, instead of directly computing the Hessian matrix, we approximate it using $L_t^\top L_t +\alpha I$ and $R_t^\top R_t +\alpha I$, which avoids the computational burden of inverting the Hessian matrix.
> > >
> > > Finally, we have corrected the issue with Figure 3 on page 11 in the latest version.

---

### Meta-Review · Area_Chair_t6uE · 2024-12-21

**Metareview:**

This paper addresses low-rank matrix sensing by proposing an alternating preconditioned gradient descent algorithm that can be applied to asymmetric matrices. The main contribution lies in generalizing previous approaches—originally restricted to symmetric matrices—to the asymmetric case. The authors provide a convergence analysis showing a linear rate of convergence that is independent of the condition number of the target matrix. Although the reviewers acknowledged the significance of the problem, they raised concerns about the theoretical results, specifically regarding the readability and clarity of the presentation. Additionally, they found the initialization assumption—requiring knowledge of the largest singular value of the unknown matrix $\mathbf{X}_*$ to be impractical in real-world scenarios.
In my view, the paper addresses a relevant problem with interesting ideas. However, it requires significant revisions, particularly in closing the gap between theory and practice regarding the initialization assumption. For instance, the authors might explore ways to relax the need for prior knowledge of the largest singular value, such as using a more flexible or randomized initialization scheme. Furthermore, the paper should be rewritten to clearly position its contributions in relation to existing work. In its current form, the presentation does not sufficiently convey the key ideas of the proposed approach. Therefore, I recommend rejecting  the paper in its current form.

**Additional Comments On Reviewer Discussion:**

The reviewers pw3m, D9xn, and EsWW all raised concerns about the readability of the paper. Reviewer pw3m requested more details regarding the role of the RIP constant in the convergence analysis; in response, the authors added a discussion during the rebuttal phase. However, pw3m maintained their score, expressing doubts about the novelty of the work. Reviewer D9xn questioned the technical soundness of the paper and highlighted a gap between theory and practice due to the initialization assumption in the main theorem, which requires knowledge of the largest singular value of the unknown matrix $\mathbf{X}_*$. The authors did not adequately address these concerns or provide convincing responses to D9xn during the rebuttal phase. Lastly, reviewer EsWW  expressed concerns regarding the positioning of the proposed work in comparison to existing approaches, and again, the authors’ responses did not persuade this reviewer to increase their score. Overall, I acknowledge the authors’ efforts to address the issues raised, but their responses ultimately were not convincing to the reviewers.

---

### Decision · Program_Chairs · 2025-01-22

Reject